# ONLINE LEARNING WITH RECENCY: ALGORITHMS FOR SLIDING-WINDOW STREAMING MULTI-ARMED BANDITS

## ABSTRACT

Motivated by the recency effect in online learning, we study algorithms for single-pass *sliding-window streaming multi-armed bandits (MABs)* in this paper. In this setting, we are given $n$ arms with unknown sub-Gaussian reward distributions and a parameter $W$. The arms arrive in a single-pass stream, and only the most recent $W$ arms are considered valid. The algorithm is required to perform pure exploration and regret minimization with *limited memory*, defined as the number of stored arms. The model is a natural extension of the streaming multi-armed bandits model (without the sliding window) that has been extensively studied in recent years. We provide a comprehensive analysis of both the pure exploration and regret minimization problems with the model. For pure exploration, we prove that finding the best arm is hard with sublinear memory while finding an *approximate* best arm admits an efficient algorithm. For regret minimization, we explore a new notion of regret and give sharp memory-regret trade-offs for any single-pass algorithms. We complement our theoretical results with experiments, demonstrating the trade-offs between sample, regret, and memory.

## 1 INTRODUCTION

The stochastic multi-armed bandits (MABs) model is a fundamental model extensively studied in machine learning (ML) and theoretical computer science (TCS). In its most common form, we are given $n$ arm with unknown sub-Gaussian reward distributions, and we can learn the instance by *sampling* from the arms. The most important problems in the model include *pure exploration*, where the goal is to identify the best or a near-optimal arm, and *regret minimization*, where the aim is to devise a sampling strategy that performs competitively against the best arm in hindsight. The multi-armed bandits model has found broad applications in experiment design and clinical trials (Robbins, 1952; Pallmann et al., 2018; Simchi-Levi & Wang, 2023), financial strategies (Shen et al., 2015; Trovò et al., 2018), information retrieval (Radlinski et al., 2008; Losada et al., 2017), algorithm design (Bouneffouf et al., 2017; Gullo et al., 2023), to name a few.

Classical algorithms for MABs often assume the entire set of $n$ is stored in the memory for repeated access. However, this assumption can be unrealistic in modern online learning and large-scale applications, where arms may arrive sequentially in a stream, and the available memory is insufficient to store all of them. To address this challenge, the work of Liau et al. (2018); Assadi & Wang (2020) introduced the *streaming* multi-armed bandits model. In this model, the arms arrive one after another in a stream, and the algorithm would ideally maintain a memory substantially smaller than the total number of arms. The maximum number of arms maintained in the memory is defined as the *space complexity* of the algorithm. The streaming MABs model has attracted considerable attention since its introduction, and a flurry of work has established near-tight trade-offs for pure exploration (Assadi & Wang, 2020; Jin et al., 2021; Maiti et al., 2021; Assadi & Wang, 2022; 2024; Karpov & Wang, 2025) and regret minimization (Liau et al., 2018; Maiti et al., 2021; Agarwal et al., 2022; Wang, 2023; He et al., 2025) in various settings.

While most work on streaming MABs targets global objectives, such as identifying the best arm overall, many applications exhibit a recency effect, where recent arms matter more. For example, movie recommendation systems must adapt quickly to shifting trends. A related motivation comes from privacy constraints: regulations and policies often mandate data deletion after limited periods. GDPR requires data retention only for the "necessary" duration (GDPR, 2016), Apple retains user data for 6 months (Apple Inc., 2021), and Google limits anonymized advertising data to 9 months (Google LLC, 2025). Alas, streaming MABs algorithms usually do not take any recency effect into consideration. For instance, the pure exploration algorithms, e.g., the ones in Assadi & Wang (2020); Jin et al. (2021); Maiti et al. (2021), may output an arm that arrives very early in the stream, which is far from being recent. Similarly, the regret minimization algorithms in Maiti et al. (2021); Wang (2023); He et al. (2025) may commit to an arm that is outside the pool of recent arms[1]. As such, the following motivating open question could be asked: *could we design efficient streaming MABs algorithms that incorporate the recency effect*?

**Sliding-window streaming multi-armed bandits.** One of the most common models that capture the recency effect is the sliding-window streaming model (Datar et al., 2002; Datar & Motwani, 2016). In a typical sliding-window stream, a total of $n$ data items (arms in the context of MABs) are arriving in a stream, and only the past $W$ items are considered valid. The sliding-window streams have been extensively studied in various contexts, including frequency estimation (Datar et al., 2002; Braverman & Ostrovsky, 2007), graph algorithms (Crouch et al., 2013; Crouch & Stubbs, 2014; Zhang et al., 2024), clustering (Braverman et al., 2016; Borassi et al., 2020; Epasto et al., 2022; Woodruff et al., 2023; Cohen-Addad et al., 2025), among others (Tao & Papadias, 2006; Zhang et al., 2016).

Inspired by the success of sliding-window streams on various problems, we define the natural notion of sliding-window streaming MABs to explore the recency effect. Here, we are given $n$ arms arriving in a (single-pass) stream, and we are additionally given a window size $W$. When the $t$-th arm arrives, the arms with the arrival orders in $[t - W + 1, t]$ are considered the *valid* set of arms at this point. We emphasize that throughout the paper, $W$ and $n$ are parameters given by the problem instance, and we cannot adjust these parameters. The algorithm is allowed to store *any* arm (not limited to the window) regardless of whether the arm is valid [2]. The central problems here are therefore the *pure exploration* and *regret minimization* in sliding-window streaming MABs.

## 1.1 OUR CONTRIBUTIONS

We give a comprehensive analysis of pure exploration and regret minimization algorithms for sliding-window streaming MABs in this paper. Our results can be summarized in Table 1.

**Pure explorations.** For pure explorations, we studied both *pure exploration*, where the goal is to return the *exact* best arm, and $\varepsilon$ exploration, where the goal is to return an arm whose mean is $\varepsilon$-close to the best. In both notions, the best arm is defined as the arm with the highest mean reward in the *sliding window*. Our main conceptual message is that finding the *exact* best arm is hard unless using $\Omega(W)$ arms of memory space, but finding the *approximation* best arm is possible with sample and space efficiency.

> **Result 1** (Informal of Theorems 1 and 2)**.** *The following statements are true for exploration in sliding-window MABs.*
>
> - *Any algorithm that finds the (exact) best arm at any step with probability at least $99/100$ in the sliding-window streaming multi-armed bandits requires $\Omega(W)$ arm memory, even with an unlimited number of arm pulls.*

---

[1]This intuitively means the algorithm incurs large regret, although the definition of regret has more nuance in such cases. See Section 1.1 and Section 2 for details.

[2]The arms outside the sliding window could still be useful in various subroutines, e.g., comparing the means.

- *There exists an algorithm that finds a (approximate) $\varepsilon$-best arm with probability at least $1 - \delta$ at any steps with $O(\frac{1}{\varepsilon})$ arm memory and $O(\frac{n}{\varepsilon^2} \log \frac{W}{\delta})$ arm pulls.*

By a standard probability boosting argument, the success probability of $99/100$ in the lower bound generalizes to *any* probability of $1/2 + \Omega(1)$. On the other hand, our results demonstrate that we can identify an approximate best arm with arbitrary constant accuracy using only $O\left(\frac{1}{\varepsilon}\right)$ memory. For constant choice of $\varepsilon$ (which is usually the case), our algorithm achieves *constant memory* for exploration.

**Regret minimization.** For *regret minimization*, a significant challenge is how to *define* regret in the sliding-window model. The most natural definition is to define the regret as the cumulative gap between $\mu^*(t, W)$ and the means of the pulled arms in each window. Here, $\mu^*(t, W)$ is the mean reward of the optimal arm in the window $W$ at time $t$. However, such a definition has a fatal issue: since the algorithm controls the number of arm pulls before the window moves, the definition of the regret becomes a function of the algorithm, which means it cannot be well-defined.

To bypass the issue, we introduce the notion of *epoch-wise* regret such that the optimal reward sequences are *independent* of the arm pulls used by the algorithm. Our notion of regret minimization is to divide the total number of arms pulls $T$ into *equal-sized epochs*. Among the $n - W + 1$ epochs (one for each window position), each epoch is allocated with $\frac{T}{n-W+1}$ arm pulls. Total regret is defined as cumulative regret across epochs, and the algorithm is required to pull arms a constrained number of times in each time window. A formal definition of our regret notion can be found in Definition 6.

We believe that the introduction of the regret notion is a significant contribution; otherwise, there is no obvious way to study regret minimization in sliding-window streaming MABs. Moreover, the epoch-wise regret definition captures many practical scenarios. For instance, in the case of movie recommendations, we treat the "sliding window" as time periods of, e.g., 1-2 months, and we aim to recommend the most relevant movies in each period. Our main conceptual finding for regret minimization is that a memory of $\Omega(W)$ arms is necessary to achieve $o(T)$ regret; furthermore, there is a sharp memory-regret transition around the $\Theta(W)$ arm memory.

**Result 2** (Informal of Theorem 3). *Any algorithm that achieves $o(T/W^2)$ regret in the* epoch-wise *regret setting requires $\Omega(W)$ arm memory. Furthermore, there exist algorithms that given a stream of $n$ arms and parameters $T$ and $W$, with $O(W)$ memory achieve $O(\sqrt{W \cdot (n - W) \cdot T})$ regret.*

In the centralized setting, the tight bound for regret minimization is $O(\sqrt{nT})$, even with unlimited memory. Since $O(\sqrt{W(n - W)T}) = O(\sqrt{nT})$ when $|n - W|$ or $W$ is small, this shows that our bound for *epoch-wise regret* setting is indeed tight in the worst case. We find the conceptual message quite interesting, and we believe it could serve as important guidelines for related applications. A variant of our regret setting is when the best arm does not expire with the movement of the sliding window. While the setting is less interesting, we do believe it has applications as well. A discussion of this setting can be found in **??**.

**Our techniques.** We start with $\varepsilon$-exploration in sliding-window streaming MABs, in which there are two main technical challenges: the memory constraints and the expiration of arms. An efficient sliding-window algorithm would imply an efficient streaming algorithm (by setting $W = n$); as such, for any MABs technique to work in the sliding-window setting, there must be a streaming algorithm as well. For the majority of MABs techniques, efficient streaming algorithms either do not exist (e.g., elimination-based algorithms Even-Dar et al. (2006); Karnin et al. (2013)), or it is unclear how to design such algorithms (e.g., non-stationary bandits Whittle (1988) and mortal bandits Chakrabarti et al. (2008)). Therefore, we have to find technical ideas from existing streaming MABs algorithms (e.g. Assadi & Wang (2020); Jin et al. (2021); Maiti et al. (2021)).

Most of the algorithms in streaming MABs are either based on amortizing the sample complexity across the stream or using a bucket-based idea group arms based on the empirical means. The idea of amortization faces

| Task | Space | Sample/Regret | Remark |
|------|-------|---------------|--------|
| Exact Exploration | $\Omega(W)$ | Any | Lower Bound |
| Strong $\varepsilon$-exploration | $O(1/\varepsilon)$ | $\Theta(\frac{n}{\varepsilon^2}\log n)$ | Upper and Lower Bounds |
| Weak $\varepsilon$-exploration | $O(1/\varepsilon)$ | $\Theta(\frac{n}{\varepsilon^2}\log W)$ | Upper Bound |
| Regret minimization | $o(W)$ | $\Omega(T/W^2)$ | Lower Bound |
| | $\Omega(W)$ | $O(\sqrt{W\cdot(n-W)\cdot T})$ | Upper Bound |

Table 1: Summary of the results for exploration and regret minimization

a barrier aimed at the expiration of arms. In particular, for the algorithms that amortize sample complexity in, e.g., Assadi & Wang (2020); Jin et al. (2021), the guarantees are only given with respect to the best arm, and the analysis falls apart if the best arm changes due to the sliding window movements. On the other hand, the grouping of empirical means based on the multiplicative of $\varepsilon$ is naturally compatible with the sliding-window model. Here, we can simply discard the expired arms, and the invariant among the buckets helps maintain $\varepsilon$-best arms. This is the main idea for our $\varepsilon$-exploration algorithms.

Our lower bound for the exact pure exploration establishes a sharp dichotomy between the exact and approximate $\varepsilon$-exploration problems. Here, the main challenge is to find a distribution that forces the algorithm to store virtually all arms. Our idea is to use a distribution of arms with *decreasing mean rewards*. This distribution forces the "useless" arm when the window is at position $t$ to become optimal when the window moves to $t+1$. Although the distribution is not involved, it crucially uses the sliding-window property to separate from the streaming case, especially given that the latter admits an efficient algorithm with a single-arm memory (Assadi & Wang (2020)). Finally, our regret lower bound follows the same idea, although we need to extend the distribution to slightly more involved ones to ensure the algorithm cannot get "lucky" with the instance distribution.

**Experiments.** We conducted experiments for both pure exploration and regret minimization applications[3]. For pure exploration, we implemented the $\varepsilon$-best pure exploration algorithm, and for regret minimization, we used the $O(W)$-memory algorithms outlined in Result 2. These are the first algorithms designed to work with multi-armed bandits (MABs) under a sliding-window setting.

In our pure exploration experiments, we tested configurations with $n \in \{1000, 2000, 5000\}$ and $n \in \{10, 20, 50\}$. The results indicate a relatively smooth trade-off between the quality of the returned arm and the memory used. The error exceeded $0.6$ in all settings when we employed a memory size of $0.05W$; however, it dropped to below $0.3$ with a memory size of $0.3W$. On the other hand, we can easily show that existing algorithms could result in $0.6$ error (**??**), and our empirical results essentially mean that with $0.3W$ memory, the error could be reduced by $50\%$. For the regret minimization experiments, we tested configurations with $n \in \{500, 1000, 2000\}$ and $n \in \{10, 20, 50\}$, while setting the number of pulls for each epoch to $\frac{T}{n-W+1} = 1000$. The results revealed sharp changes in regret around the memory size $W$, confirming our theoretical predictions. The total regret decreased by more than $50\%$ for most configurations when the memory size increased from $0.05W$ to $W$.

---

[3]Our code is available on anonymous Github: `https://anonymous.4open.science/r/sliding-window-MABs-CF74/`.

## 2 Problem Definition and Preliminaries

In this section, we give the formal definition of the problems we investigated and some standard technical tools. We start with a formal definition of stochastic MABs.

**Definition 1** (Stochastic multi-armed bandits (MABs) model). In the stochastic multi-armed bandits model, we have a collection of $n$ arms $\{\text{arm}_i\}_{i=1}^n$, and each arm follows a distribution with mean $\mu_i \in [0, 1]$. Each pull of $\text{arm}_i$ returns a sample from the distribution with mean $\mu_i$.

Note that by the central limit theorem, sampling from arbitrary distributions over $[0, 1]$ is essentially the same as sampling from an arbitrary sub-Gaussian distribution (up to a scaling factor). The sliding-window streaming MABs could therefore be defined as follows.

**Definition 2** (The sliding-window streaming MABs model.). In the sliding-window streaming MABs model, we have a collection of $n$ arms $\{\text{arm}_i\}_{i=1}^n$ arranged in order and a window size $W^4$. Each arm follows a distribution with mean $\mu_i \in [0, 1]$. The arms arrive one by one in the stream, and we let $\{\text{arm}_i\}_{i=t-W+1}^t$ be the set of valid arms that arrived in the $W$ latest steps. When a new arm arrives, the algorithm can pull the arriving arm and the arms in memory. The algorithm can also decide whether to store the new arm in memory or discard it, and the algorithm can discard some arms stored in memory to free up space. At any point, the collection of arms that the algorithm could access are the arms in memory and the arriving arm.

**Remark 1.** To keep consistent with the literature in sliding-window streaming algorithms, e.g., Datar et al. (2002); Datar & Motwani (2016), we do *not* force the algorithm to discard the expired arms. Nevertheless, our upper and lower bounds in Result 1 and Result 2 do *not* rely on this property. In other words, if we add the condition that the expired arms have to be deleted immediately, the upper and lower bounds in Result 1 and Result 2 still hold. The immediate removal of expired arms from the memory is helpful for applications with private data retention requirements.

We can now define the *sample and space complexity* of a sliding-window streaming MABs algorithm.

**Definition 3** (*Sample complexity*). The *sample complexity* of a sliding-window streaming MABs algorithm is defined as the total number of pulls of the algorithm.

**Definition 4** (*Space complexity*). The *space complexity* of a sliding-window streaming algorithm is defined as the maximum number of arms that we store in the memory at any time during the algorithm.

**Pure exploration.** One of the most natural problems in the MABs problem in the sliding-window model is the *pure exploration* problem, where the algorithm is asked to return the best or near-best arms. In what follows, we discuss the necessary notions before formally defining the pure exploration problems.

**Definition 5** (*Best arm in the window*). Assume that we have a collection of $n$ arms $\{\text{arm}_i\}_{i=1}^n$ with means $\mu_i$ and arranged in the streaming arriving ordered. Let $W$ be the window size and $t$ be the index of the current arriving arm. Then, for any $t \in [n]$, the best arm in the window $\text{arm}^*(W, t)$ is the arm with the highest mean $\mu^*(W, t)$ among the $W$ latest arms $\{\text{arm}_i\}_{i=t-W+1}^t$.

Note that the notation $\text{arm}^*(W, t)$ is a function of $t$ and $W$. We also call the set of arms $\{\text{arm}_i\}_{i=t-W+1}^t$ *valid* at time step $t$ for fixed $t$ and $W$.

We are ready to introduce the *pure exploration* problem for the sliding-window streaming MABs model.

**Problem 1** (Exact pure exploration in sliding-window MABs). Given a stream of $n$ arms $\{\text{arm}_i\}_{i=1}^n$ and a window size $W$, we say a sliding-window streaming MABs algorithm ALG solves

- *weak* pure exploration with probability $1 - \delta$ if at any time $t \in [n]$, ALG can output the best arm in the window with probability at least $1 - \delta$.

---

[4]We emphasize that the parameter $W$ is an input parameter (not the algorithm's choice).

- *strong* pure exploration with probability $1 - \delta$ if ALG can output the best arm in the window at all time $t \in [n]$ with probability $1 - \delta$.

Next, we could analogously define the $\varepsilon$ exploration problem in both the *weak* and the *strong* versions for the sliding-window streaming MABs.

**Problem 2** ($\varepsilon$ exploration in sliding-window MABs). Given a stream of $n$ arms $\{\mathtt{arm}_i\}_{i=1}^n$, a window size $W$, and a parameter $\varepsilon$, we say a sliding-window streaming MABs algorithm ALG solves

- *weak $\varepsilon$ exploration* with probability $1 - \delta$ if at any time $t \in [n]$, ALG is able to output an arm with mean reward $\mu$ such that $\mu \geqslant \mu^*(t, W) - \varepsilon$ with probability at least $1 - \delta$.
- *strong $\varepsilon$ exploration* with probability $1 - \delta$ if ALG is able to output an arm with mean reward $\mu$ such that $\mu \geqslant \mu^*(t, W) - \varepsilon$ at all time $t \in [n]$ with probability $1 - \delta$.

Here, as defined in Definition 5, $\mu^*(t, W)$ is the mean reward of the best arm in the window.

**Regret minimization.** In Section 1.1, we have discussed the high-level definition for our regret notion in sliding windows, i.e., the epoch-wise regret. We now introduce the formal definition as follow.

**Definition 6** (Regret minimization with epoch-wise regrets). Let $\{\mathtt{arm}_i\}_{i=1}^n$ be a collection of $n$ arms, and let $W$ and $T$ be the window size and the total number of trials. We divide $T$ into $(n - W + 1)$ equal-sized *epochs* with $\frac{T}{n-W+1}$ in each epoch. Let $t$ be the variable for the index of the arriving arm, and for any $t$, the algorithm is required to conduct *exactly* $\frac{T}{n-W+1}$ arm pulls among $\{\mathtt{arm}_i\}_{i=t-W+1}^t$. We define the regret of the $j$-th epoch as $R^E(j) = \sum_{\tau=1}^{T/(n-W+1)} (\mathtt{arm}^*(W, t) - \mathtt{arm}_{i(\tau)})$, where $i(\tau)$ is the arm index pulled by the algorithm. The total regret is defined as $R_T = \sum_{j=1}^{T/(n-W+1)} R^E(j)$, i.e., the regret over the epochs.

## 3    A LOWER BOUND FOR PURE EXPLORATION IN SLIDING-WINDOW MABS

The most natural pure exploration problem is *pure exploration* which asks to return the *best arm*. In the vanilla streaming multi-armed bandits (MABs) model, pure exploration can be solved with $O(n/\Delta_{[2]}^2)$ samples and a single-arm memory, where $\Delta_{[2]}$ represents the difference between the mean of the best and the second-best arms. As such, one would naturally wonder whether the same story applies to the sliding-window model. In this section, we will show that pure exploration is surprisingly much harder in the sliding-window streams: unless the algorithm uses $\Omega(W)$ space, we cannot obtain any algorithm that solves pure exploration.

The hard instance for our lower bound is a stream with descending mean rewards of arms, i.e., $\mu_1 > \mu_2 > \cdots > \mu_n$ for arms . The optimal solution for the sliding-window MABs would be to select $\mathtt{arm}_{n-W+1}$, which is the oldest non-expired arm. However, to always keep the oldest arm that has not expired in the memory, we would naturally need $W$ memory. The following theorem formalizes the above intuitions.

**Theorem 1.** *Any algorithm that given $n$ arms in a sliding-window stream with a window size of $W$, solves* weak or strong *pure exploration problem in sliding-window streaming multi-armed bandits with a probability of at least $99/100$ has a space complexity of at least $\Omega(W)$, even if the sample complexity is unbounded.*

*Proof.* We prove the theorem for weak pure exploration, since the task of strong pure exploration is only harder. In other words, since the answer for strong exploration is always valid for weak exploration, the former task should use at least the same amount of memory and samples.

By Yao's minimax principle (Yao, 1977), it is sufficient to prove the lower bound for deterministic algorithms over a challenging distribution of inputs. Let $n = 2W$. We construct the instance $\{\mathtt{arm}_1\}_{i=1}^n$ such that $\mu_i = 1 - \frac{i}{3W}$. To solve the *weak* pure exploration problem with a probability of at least $\frac{99}{100}$, the algorithm must correctly identify at least $\frac{49}{50}$ of the best arms in the second half of the stream $\{\mathtt{arm}_i\}_{i=1}^n$. If the algorithm fails to do this, the overall success probability would drop below $1 \cdot \frac{1}{2} + \frac{49}{50} \cdot \frac{1}{2} = \frac{99}{100}$.

Let $T \subset \{W+1, W+2, \ldots, 2W\}$ represent the collection of times when the algorithm correctly identifies the best arm in the window during the second half of the stream. Define $A = \{\texttt{arm}^*(W,t)|t \in T\}$ as the set of best arms in the window at times $t \in T$. For any $t \in \{W+1, W+2, \ldots, 2W\}$, the best arm in the window $\texttt{arm}^*(W,t)$ should be $\texttt{arm}_{t-W+1}$ because the expected values of the arms monotonically decrease in this instance. Therefore, we have $A = \{\texttt{arm}_{t-W+1}|t \in T\}$. Given that $T \subset \{W+1, W+2, \ldots, 2W\}$ and $|T| \geqslant \frac{49}{50}W$, it follows that $A \subset \{\texttt{arm}_2, \texttt{arm}_3, \ldots, \texttt{arm}_{W+1}\}$ and $|A| = |T| \geqslant \frac{49}{50}W$.

For any $W+1 \leqslant t < 2W$, $\texttt{arm}^*(W,t) = \texttt{arm}_{t-W+1}$ has already arrived by time $W+1$. Therefore, for any $t \in T \cap [2W-1]$, $\texttt{arm}^*(W,t)$ must be stored in memory by time $W+1$ so that it can be returned at time $t$. This means that at least $|A| - 1 = \frac{49}{50}W - 1$ arms must be stored in memory at time $W+1$. Hence, according to Yao's minimax principle, the algorithm must have a *space complexity* of at least $\Omega(W)$. $\qquad \square$

Note that the success probability of $99/100$ in the theorem is not inherently special: by a simple probability boosting argument, we can always maintain $O(1)$ copies of the algorithm and output the majority with asymptotically the same memory and number of samples. As such, our lower bound in Theorem 1 applies to any success probability of $1/2 + \Omega(1)$.

## 4 SLIDING-WINDOW ALGORITHMS AND LOWER BOUNDS FOR $\varepsilon$-PURE EXPLORATION

Section 3 depicts a very pessimistic picture for the pure exploration of the *best arm* in sliding-window streaming MABs. A natural question to follow is whether we could get positive results using a relaxed notion. A natural candidate for this purpose is the $\varepsilon$ exploration under the $(\varepsilon, \delta)$-PAC framework. Here, instead of returning the *single best* arm, we are allowed to obtain an arm whose gap is within $\varepsilon$ additive to the best, i.e., return an arm with mean reward $\mu \geqslant \mu^* - \varepsilon$. In this section, we present the bounds for both *strong* and *weak* $\varepsilon$ exploration. Our main results are:

- A pure exploration algorithm that solves *weak $\varepsilon$* exploration with probability $1 - \delta$ in the sliding-window streaming MABs model with $O\left(\frac{n}{\varepsilon^2} \log \frac{W}{\delta}\right)$ *sample complexity* and $O\left(\frac{1}{\varepsilon}\right)$ *space complexity*.
- A lower bound shows that for any algorithm to solve *strong $\varepsilon$* exploration with probability $99/100$ in the sliding-window streaming MABs model, the algorithm has to use $\Omega(\frac{n}{\varepsilon^2} \log \frac{n}{W})$ sample complexity. Since $n \gg W$ in most cases, the lower bound separated the *weak* and *strong $\varepsilon$* exploration problems in the sliding-window streaming MABs model.
- Finally, we give a nearly-matching algorithm for *strong $\varepsilon$* exploration with probability $1 - \delta$ in the sliding-window streaming MABs model with $O\left(\frac{n}{\varepsilon^2} \log \frac{n}{\delta}\right)$ *sample complexity* and $O\left(\frac{1}{\varepsilon}\right)$ *space complexity*.

### 4.1 AN EFFICIENT ALGORITHM FOR WEAK $\varepsilon$-PURE EXPLORATION

We start with introducing a streaming algorithm designed for *weak $\varepsilon$* exploration.

**Theorem 2.** *There exists a streaming algorithm that, given $n$ arms arriving in a sliding-window stream with a window size $W$ and a confidence parameter $\delta$, solves* weak $\varepsilon$ *exploration with a probability of at least $1 - \delta$ using a sample complexity of $O\left(\frac{n}{\varepsilon^2} \log \frac{W}{\delta}\right)$ and a space complexity of $O\left(\frac{1}{\varepsilon}\right)$.*

At a high level, the algorithm follows the idea of partitioning the range $[0, 1]$ into $O\left(\frac{1}{\varepsilon}\right)$ segments ("buckets") of equal length. An arm is considered to belong to a bucket if its mean value falls within the range of that segment. For an arm $\texttt{arm}_i$ that belongs to bucket $B$, any arm $\texttt{arm}'$ that is in a nearby bucket would serve as an $\varepsilon$-approximation of $\texttt{arm}_i$. If we pull each arm an adequate number of times, we can ensure that any arm is placed into a bucket that is close enough to its mean; thus, the non-expired arm from the highest bucket will be an $\varepsilon$-best arm. To optimize memory usage, we store only the latest arm for each bucket instead of all the arms that belong to that bucket. Our algorithm for *weak $\varepsilon$* exploration is presented in Algorithm 1, with the pulling size set to $s = (9/2\varepsilon^2) \cdot \ln 6W/\delta$.

---

**Algorithm 1:** Efficient Algorithm for $\varepsilon$ exploration in Sliding-window Streaming MABs: BUCKET($s$)

---

**Input:** Data stream $\{\text{arm}_i\}_{i=1}^n$, window size $W$, *confidence parameter* $\delta$ and *accuracy parameter* $\varepsilon$;

**Input:** Sample complexity: $s = \frac{9}{2\varepsilon^2} \ln \frac{6W}{\delta}$ for weak exploration and $s = \frac{9}{2\varepsilon^2} \ln \frac{6n}{\delta}$ for strong exploration;

**Output:** $\varepsilon$-best arms $\{\widehat{\text{arm}}_i\}_{i=1}^n$;

$N \leftarrow \frac{3}{\varepsilon}$;

Generate $N$ buckets $B_1, B_2, \cdots, B_N$;

**for** *each arriving arm* $\text{arm}_i$ **do**

    Pull $\text{arm}_i$ for $s$ times and evaluate empirical mean $\widehat{\mu}_i$;

    Store $\text{arm}_i$ in $B_j$ such that $(j-1)\frac{\varepsilon}{3} < \widehat{\mu}_i \leqslant j\frac{\varepsilon}{3}$ and discard the arms stored in $B_j$ previously;

    Discard all stored arms that are expired;

    $\widehat{\text{arm}}_i \leftarrow$ the arm stored in $B_k$ such that $k = \max_{i \leqslant N}\{B_i \neq \emptyset\}$

**end**

**return** $\{\widehat{\text{arm}}_i\}_{i=1}^n$

---

## 4.2 A LOWER BOUND FOR STRONG $\varepsilon$-PURE EXPLORATION

We will now discuss the lower bound for *strong $\varepsilon$ exploration* that has an extra $\log n$ factor. In particular, if we show that when $W \ll n$ (e.g., $W = \log n$), there is a lower bound of $\Omega(\frac{n}{\varepsilon^2} \log n)$ samples for strong exploration, it would imply a *separation* between the *weak* and *strong $\varepsilon$ exploration* since the weak exploration only requires $\Omega(\frac{n}{\varepsilon^2} \log W)$ samples by Algorithm 1 BUCKET $\left(\frac{9}{2\varepsilon^2} \ln \frac{6W}{\delta}\right)$. Then we have:

**Lemma 4.1.** *For infinitely many choices of parameters $n$, $\varepsilon$, and $W \leqslant n^{0.99}$, there exists a distribution of arms $\mathcal{D}(n, W, \varepsilon)$ such that any algorithm that solves the* strong $\varepsilon$ exploration *with probability at least $99/100$ on $\mathcal{D}(n, W, \varepsilon)$ requires at least $\Omega(\frac{n}{\varepsilon^2} \log n)$ samples. The lower bound holds even if the algorithm is with unbounded memory.*

The technical statement for Lemma 4.1 is more general and gives $\Omega(\frac{n}{\varepsilon^2} \log \frac{n}{W})$ samples for $W \in [1, n/8]$, although the bound is less informative when $W$ is large. At a high level, our lower bound works by reducing solving *independent* copies of the $\varepsilon$-best arm identification to the sliding-window streaming $\varepsilon$ exploration case. Mannor & Tsitsiklis (2004) proved that $O\left(\frac{n}{\varepsilon^2} \log\left(\frac{1}{\delta}\right)\right)$ pulls are necessary to identify an $\varepsilon$-best arm among $n$ arms with a probability of at least $1 - \delta$.

In the slide-window setting, since arms will expire after $W$ time, the information from one window does not affect another *disjoint* window. There are $\Theta(\frac{n}{W})$ windows in a sliding-window stream that are disjoint. Since each window requires at least $O\left(\frac{W}{\varepsilon^2} \log\left(\frac{n}{W}\right)\right)$ pulls to solve its exploitation with a probability of at least $1 - \Theta\left(\frac{W}{n}\right)$, it follows that $O\left(\frac{n}{\varepsilon^2} \log\left(\frac{n}{W}\right)\right)$ pulls are necessary to achieve *strong $\varepsilon$ exploration* with a probability of at least $99/100$.

## 4.3 AN EFFICIENT ALGORITHM FOR STRONG $\varepsilon$-PURE EXPLORATION

We introduce a streaming algorithm for *strong $\varepsilon$ exploration*. The algorithm uses essentially the same subroutine as in Algorithm 1, but it uses a larger pulling size of $s = \frac{9}{2\varepsilon^2} \ln \frac{6n}{\delta}$ to beat a union bound.

**Lemma 4.2.** *There exists a streaming algorithm that, given $n$ arms arriving in a sliding-window stream with a window size $W$ and a confidence parameter $\delta$, solves* strong $\varepsilon$ exploration *with a probability of at least $1 - \delta$. This algorithm achieves a sample complexity of $O\left(\frac{n}{\varepsilon^2} \log \frac{n}{\delta}\right)$ and a space complexity of $O\left(\frac{1}{\varepsilon}\right)$.*

## 5 REGRET MINIMIZATION IN SLIDING-WINDOW STREAMING MABS

In this section, we investigate *regret minimization* for sliding-window streaming multi-armed bandits (MABs). Recall that in Definition 6, we defined regret minimization with the concepts of *epoch-wise* regret. Here, we have $n - W + 1$ equal-sized epochs, and we must perform $\frac{T}{n-W+1}$ pulls in each epoch. The question is how to minimize the cumulative regret over the entire horizon $[T]$.

The most natural idea is to adapt strategies in streaming MABs, e.g., (Wang, 2023), to get a low regret algorithm. In particular, when a new arm arrives, we can use Algorithm 1 to pull the arm $O(\frac{1}{\varepsilon^2}\log n)$ times and place it in the bucket. By the guarantees of Algorithm 1, we will be able to get $\varepsilon$-best arms at any step with high probability. This strategy incurs a regret of $O(\frac{1}{\varepsilon^2}\log n)$ when identifying the $\varepsilon$-best arm during each epoch. Additionally, there is a regret of $O(\varepsilon\frac{T}{n-W+1})$ for the remaining pulls on the $\varepsilon$-best arm we identify within each epoch. As a result, the total regret is $O(\frac{n}{\varepsilon^2}\log n + \varepsilon T)$. The regret is minimized by choosing $\varepsilon = O(\sqrt[3]{\frac{n\log n}{T}})$, which gives a total regret of $O(T^{\frac{2}{3}}(n\log n)^{\frac{1}{3}})$.

Alas, this strategy has a fatal issue: Algorithm 1 requires $O\left(\frac{1}{\varepsilon}\right)$ memory space; and since in most cases $T \gg n \gg W$, the memory of $1/\varepsilon = O(\sqrt[3]{\frac{T}{n\log n}})$ could be way bigger than the window size $W$. Thus, it is not immediately clear whether we could get low-regret algorithms with small memory in this setting. In this section, we show that the issue of the aforementioned algorithm is not an artifact: we prove a strong lower bound showing that a total regret of $O\left(\frac{T}{W^2}\right)$ is unavoidable if we only have $o(W)$ space.

**Theorem 3.** *There exists a family of streaming stochastic multi-armed bandit instances such that, for any given parameters $T$, $n$, and $W$, where $T \geqslant n \geqslant 16W$, any single-pass streaming algorithm for a sliding-window stream of length $n$ with a window size $W$ and a memory capacity of $\frac{W-1}{2}$ arms must incur a total expected regret given by $\mathbb{E}\left[R_T\right] \geqslant \frac{T}{64W^2}$.*

*Furthermore, there exists an algorithm that given $n$ arms arriving in a stream and parameters $W$ and $T$, achieves $O(\sqrt{W \cdot (n - W) \cdot T})$ total regret with $W$ memory.*

At a high level, our lower bound is obtained by constructing $W$ arms whose means decrease by $\frac{1}{W}$ and $W$ arms with the same mean, and the pattern is repeated over the stream. Since we can only store at most half of these arms, if the best arm in the epoch is missed, the regret for each pull will be at least $\frac{1}{2W}$. This leads to a total regret of $\Omega\left(\frac{T}{W^2}\right)$. Our upper bound is obtained by running UCB-based algorithms on each window.

## 6 EXPERIMENTS

We conduct experiments for both $\varepsilon$-exploration and regret minimization in the sliding-window streaming setting. Our main empirical finding is that, consistent with our theoretical results, both the $\varepsilon$-exploration and regret minimization algorithms demonstrate trade-offs between memory and quality/regret. The regret minimization algorithm demonstrates a sharp change around the $O(W)$-arm memory. We will briefly demonstrate the experiments of the $\varepsilon$-exploration and regret minimization algorithms in epoch-wise settings. Additional experimental results can be found in **??**.

**The data.** We use synthetic data with streams of arms to conduct our experiments. We use different types of instances for exploration and regret minimization as follows.

- For exploration, we sample $n$ arms with the distribution $\mathsf{Bern}(p)$ such that $p$ is from a uniform distribution[5]. We note that the "uniform" type of instances are more suitable for $\varepsilon$-exploration since the quality decrement

---

[5]We use $\mathsf{Bern}(p)$ to denote Bernoulli distribution with mean $p$.

of the returned arms could be better captured. We use $n \in \{1000, 2000, 5000\}$ and $W \in \{10, 20, 50\}$ for $\varepsilon$-exploration experiments.

- For regret minimization, we need instance distributions *consistent* with our instance distribution in Section 5. For the epoch-wise regret minimization, we sample $n - n/W$ arms with distribution $\mathsf{Bern}(0.25)$ and $n/W$ arms with distribution $\mathsf{Bern}(0.95)$. We then permute the arms uniformly. Due to constraints on running time, we use $n \in \{500, 1000, 2000\}$ and $W \in \{10, 20, 50\}$ for $\varepsilon$-exploration experiments.

To mitigate the noise from randomness, for each parameter setting with fixed memory size, we conduct 10 **independent runs of experiments and take the average**. For the quality of the arm and the regret minimization, we also report error bars and the ranges of the regrets.

**The algorithms.** We conduct experiments for both exploration and regret minimization. We first implement and test our $\varepsilon$-exploration algorithm (Algorithm 1). Note that it is hard to compare performances with baseline algorithms for sliding-window exploration since we cannot easily quantify "error" when the algorithm outputs an expired arm. Therefore, we report the performance of our algorithm with different memory sizes.

For regret minimization, we adapt the algorithm with $W$-arm memory discussed in Section 5. To handle the case of $m < W$-arm memory, we simulate the reservoir sampling: after the memory is full, for each arriving arm, we toss a fair coin with bias $m/t$ for the $t$-th arriving arm to decide whether we admit the new arm to the memory (by uniformly at random discarding an arm existing in the memory).

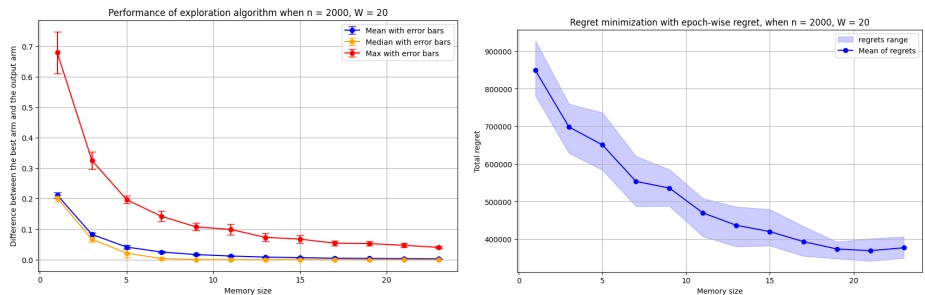

Figure 1: The performances of $\varepsilon$-exploration and regret minimization, $n = 2000$, $W = 20$.

**Summary of the experiments.** A sample of the performances for $\varepsilon$-exploration and regret minimization is given in Figure 1 (for $n = 2000$ and $W = 20$; see **??** for more parameter settings). As we can observe from the figures, for all the experiments, there is generally a trade-off between the arm quality/regret and memory. The trade-off in $\varepsilon$-exploration is generally smoother, and the regret minimization for the everlasting best arm demonstrates a sharp drop of regret around the $W$-memory point. These results are consistent with our theoretical findings for sliding-window streaming MABs algorithms.

## 7 CONCLUSION AND FUTURE WORK

In this work, we initiated the study of multi-armed bandits (MABs) in the sliding-window model. Our results built the fundamental hardness of online learning in the sliding-window MABs model, and we provided important insights for related applications, e.g., using $\varepsilon$-exploration rather than pure exploration in practice. There are several open directions to follow up on our work. For instance, one appealing question is the *multi-pass* setting: if the algorithm is allowed to make multiple passes over the stream, it might be possible for the algorithm to achieve better memory efficiency. The sliding-window model for other variants of MABs, e.g., the linear bandits, can be another interesting direction to pursue.

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
