## A   TECHNICAL PRELIMINARIES

We give some technical preliminaries of our paper in this section.

**Concentration inequalities.**    We use some standard concentration inequalities in the proof of our results. We provide these inequalities for completeness.

**Proposition A.1** (Chernoff-Hoeffding bound)**.** *Let $X_1, X_2, \cdots, X_m$ be a sequence of independent discrete random variables bounded in the range $[0, 1]$. Define $S_m = \sum_{i=1}^m X_i$, then*

$$\mathbf{Pr}\left[|S_m - \mathbb{E}[S_m]| \geqslant t\right] \leqslant 2 \cdot \exp\left(-\frac{2t^2}{m}\right).$$

We also use the following direct corollaries of the Chernoff-Hoeffding bound.

**Proposition A.2.** *Let* `arm` *be an arms with mean $\mu$. We pull the arm $\frac{K}{\theta^2}$ times to obtain empirical mean $\widehat{\mu}$. Then,*
$$\mathbf{Pr}\left[|\mu - \widehat{\mu}| \geqslant \theta\right] \leqslant 2 \cdot \exp\left(-2K\right).$$

**Proposition A.3.** *Let* `arm`$_1$ *and* `arm`$_2$ *be two different arms with means $\mu_1$ and $\mu_2$. Suppose $\mu_1 - \mu_2 \geqslant \theta > 0$ and we pull each arm $\frac{K}{\theta^2}$ times to obtain empirical rewards $\widehat{\mu}_1$ and $\widehat{\mu}_2$. Then,*

$$\mathbf{Pr}\left[\widehat{\mu}_1 \leqslant \widehat{\mu}_2\right] \leqslant 2 \cdot \exp\left(-\frac{K}{4}\right).$$

## B   ADDITIONAL RELATED WORK

Apart from the streaming MABs algorithms, our work is also closely related to settings with evolving and expiring arms, including arm-acquiring bandits Whittle (1981), non-stationary bandits Whittle (1988), mortal bandits Chakrabarti et al. (2008), and the sleeping expert problems Kleinberg et al. (2010). For instance, in the mortal bandits problem, the arm can expire after a certain number of samples, or it goes expiring with some probability at each step. We remark that although these settings are in a similar spirit of arm expiration, their settings are not directly comparable with ours since we always prioritize the most recent arms in the sliding-window model. Furthermore, none of these problems considered *memory* constraints, which means their algorithms cannot be directly applied in our setting.

## C   THE COMPLETE DETAILS FOR RESULTS IN SECTION 4

In this section, we provide the complete details (missing algorithms and analysis) we discussed in Section 4.

### C.1   THE ANALYSIS OF THEOREM 2 AND ALGORITHM 1

We now proceed to the analysis of Algorithm 1. The following lemma establishes the *space complexity* of the algorithm.

**Lemma C.1.** *The space complexity of* BUCKET $\left(\frac{9}{2\varepsilon^2} \ln \frac{6W}{\delta}\right)$ *is $O\left(\frac{1}{\varepsilon}\right)$.*

*Proof.* Since we discard the previous arm when storing a new arm in a bucket, each bucket will contain at most one arm during the execution of the algorithm. Therefore, the space complexity of the algorithm is bounded by $N = \frac{3}{\varepsilon}$, which is $O\left(\frac{1}{\varepsilon}\right)$. $\qquad\square$

The following lemma provides a bound on the sample complexity of the algorithm.

**Lemma C.2.** *The sample complexity of* BUCKET $\left(\frac{9}{2\varepsilon^2} \ln \frac{6W}{\delta}\right)$ *is* $O\left(\frac{n}{\varepsilon^2} \log \frac{W}{\delta}\right)$.

*Proof.* Since we sample each arm $l = \frac{9}{2\varepsilon^2} \ln \frac{6W}{\delta}$ times within the algorithm, the *sample complexity* is given by $n \cdot l = O\left(\frac{n}{\varepsilon^2} \log \frac{W}{\delta}\right)$. $\qquad \square$

Finally, we prove the correctness of the algorithm.

**Lemma C.3.** *At any time* $t \in [n]$*, the arm* $\widehat{\text{arm}}_t$ *outputted by the algorithm* BUCKET $\left(\frac{9}{2\varepsilon^2} \ln \frac{6W}{\delta}\right)$ *is an* $\varepsilon$-*best arm for* $\text{arm}^*(t, W)$ *with a probability of at least* $1 - \delta$.

*Proof.* At any time $t \in [n]$, let $\text{arm}^*(t, W) = \text{arm}_k$ and $\widehat{\text{arm}}_t = \text{arm}_m$. Additionally, let $b_i$ be the index of the correct bucket to which $\text{arm}_i$ belongs; that is, $\mu_i \in (b_i - 1)\frac{\varepsilon}{3}, b_i \frac{\varepsilon}{3}]$. We also define $b'_i$ as the index of the bucket where $\text{arm}_i$ is stored upon arrival; thus, $\widehat{\mu}_i \in ((b'_i - 1)\frac{\varepsilon}{3}, b'_i \frac{\varepsilon}{3}]$.

Let $\mathcal{E}_i$ be the event that $|b_i - b'_i| \leqslant 1$, indicating that $\text{arm}_i$ is stored in a bucket close to its correct bucket. By Lemma A.2, we have

$$\mathbf{Pr}\left[\neg \mathcal{E}_i\right] = \mathbf{Pr}\left[|b_i - b'_i| > 1\right]$$
$$\leqslant \mathbf{Pr}\left[|\widehat{\mu}_i - \mu_i| > \frac{\varepsilon}{3}\right]$$
$$\leqslant 2 \cdot \exp\left(-2l \cdot \frac{\varepsilon^2}{9}\right) = \frac{\delta}{3W}.$$

The output $\widehat{\text{arm}}_t$ must be an $\varepsilon$-best arm if the following events occur at time $t$:

- $\mathcal{F}_1$: There exists some arms stored in $B_{b_k-1} \cup B_{b_k} \cup B_{b_k+1}$;

- $\mathcal{F}_2$: All the bucket $B_j$ for $j > b_k + 1$ are empty;

- $\mathcal{F}_3$: $|b_m - b'_m| \leqslant 1$.

We will output an arm from $\{B_{b_k-1}, B_{b_k}, B_{b_k+1}\}$ if both $\mathcal{F}_1$ and $\mathcal{F}_2$ hold. $\mathcal{F}_3$ guarantees that the output $\widehat{\text{arm}}_t$ is stored in a bucket close to its correct bucket. If all three events occur simultaneously, we have $b'_m \in \{b_k - 1, b_k, b_k + 1\}$ and $|b'_m - b_m| \leqslant 1$. Therefore, $|b_m - b_k| \leqslant 2$, leading to $|\mu_k - \mu_m| \leqslant 3 \cdot \frac{\varepsilon}{3} = \varepsilon$. This means that $\widehat{\text{arm}}_t$ is an $\varepsilon$-approximation of $\text{arm}^*(t, W)$.

We can analyze the probabilities of each event:

$\mathbf{Pr}\left[\mathcal{F}_1\right] \geqslant \mathbf{Pr}\left[\mathcal{E}_k\right] \geqslant 1 - \frac{\delta}{3W}$. This is because if $\mathcal{E}_k$ occurs, we will store $\text{arm}^*(t, W) = \text{arm}_k$ in bucket $B_{b'_k}$, where $|b'_k - b_k| \leqslant 1$. Since $\text{arm}^*(t, W)$ is the best arm at time $t$, it cannot expire, implying we will not drop the arm stored in $B_{b'_k}$ due to expiration. Thus, $B_{b'_k}$ remains non-empty.

$\mathbf{Pr}\left[\mathcal{F}_2\right] \geqslant \mathbf{Pr}\left[\cup_{j=t-W+1}^{t}\mathcal{E}_t\right] \geqslant 1 - W \cdot \frac{\delta}{3W}$. If all $\mathcal{E}_j, j \in [t - W + 1, t]$ occur, each arm will be stored in a bucket near its correct bucket. Since $\text{arm}^*(t, W) = \text{arm}_k$ is the best arm at time $t$, we have $b_j \leqslant b_k$ for any $j \in [t - W + 1, t]$. Therefore, $b'_j \leqslant b_k + 1$ for any $j$ when $\mathcal{E}_j$ occurs, thus ensuring all buckets $B_j$ for $j > b_k + 1$ are empty.

$\mathcal{F}_3$ is simply the same event as $\mathcal{E}_m$, so $\mathbf{Pr}\left[\mathcal{F}_3\right] \geqslant 1 - \frac{\delta}{3W}$.

Consequently, we obtain:

$$\mathbf{Pr}\left[\mathcal{F}_1 \cap \mathcal{F}_2 \cap \mathcal{F}_3\right] \geqslant 1 - (2 + W) \cdot \frac{\delta}{3W} \geqslant 1 - \delta.$$

$\qquad \square$

## C.2 THE ANALYSIS OF LEMMA 4.1

We now prove Lemma 4.1 with the general $\Omega(\frac{n}{\varepsilon^2} \cdot \log \frac{n}{W})$ sample lower bound (this directly implies the lower bound when $W \leqslant n^{0.99}$). We will employ the *sample complexity* lower bound established by Mannor & Tsitsiklis (2004) to prove our lemma. We provide the proposition for completeness.

**Proposition C.4** (Mannor & Tsitsiklis (2004)). *There exist positive constants $c_1, c_2, \varepsilon_0$ and $\delta_0$, such that for every $n \geqslant 2$, $\varepsilon \in (0, \varepsilon_0)$ and $\delta \in (0, \delta_0)$, and for every algorithm outputs $\varepsilon$-best arm with probability at least $1 - \delta$, there exists some $\mu = (\mu_1, \mu_2, \cdots, \mu_n) \in [0, 1]^n$ such that*

$$\mathbb{E}_\mu [T] \geqslant c_1 \frac{n}{\varepsilon^2} \log \left( \frac{c_2}{\delta} \right).$$

*$T$ is the number of pulls used in the algorithm. $\mathbb{E}_\mu [T]$ is the expectation of $T$ when arms have means $\mu = (\mu_1, \mu_2, \cdots, \mu_n)$.*

*In particular, $\varepsilon_0$ and $\delta_0$ can be taken equal to $1/8$ and $\frac{e^{-4}}{4}$, respectively.*

We assert that any algorithm capable of solving the *strong $\varepsilon$* exploration for a stream of $n$ arms, with a window size $W$, and achieving a success probability of at least $99/100$, modified to create an algorithm that addresses $\Theta\left(\frac{n}{W}\right)$ independent $\varepsilon$ exploration of $W$ arms concurrently, also with a success probability of at least $99/100$. Moreover, the *sample complexity* of the latter algorithm will be less than or equal to the *sample complexity* of the former algorithm.

Specifically, consider sets $X_i$, for $i \in \left[\frac{n}{2W}\right]$, are $\frac{n}{2W}$ sets, where each set $X_i$ consists of $W$ arms. Let $Z$ denote a set that contains $W$ arms, each of which consistently returns 0. We can construct a stream $S = (Z, X_1, Z, X_2, \cdots, Z, X_{\frac{n}{2W}})$. By employing an algorithm designed to solve the *strong $\varepsilon$* exploration task on the stream $S$, we can simultaneously solve the $\varepsilon$ exploration problem for all sets $X_i$.

**Lemma C.5.** *If an algorithm* ALG *exists that successfully solves the* strong $\varepsilon$ *exploration problem for a stream of $n$ arms with a window size $W$ with a probability of at least $99/100$, and has a sample complexity $m$, then there exists another algorithm* ALG$'$ *that can solve $\frac{n}{2W}$ independent $\varepsilon$ exploration problems for $W$ arms simultaneously. This new algorithm* ALG$'$ *will have a sample complexity $m'$ such that $m' \leqslant m$, while also achieving a success probability of at least $99/100$.*

*Proof.* We demonstrate the lemma by providing a framework, Algorithm 2, which generates the algorithm ALG$'$ based on the algorithm ALG.

By the construction of $S$, an $\varepsilon$-best arm at time $i \cdot 2W$ must also be an $\varepsilon$-best arm among the set $\{\widehat{\mathrm{arm}}_{i,j}\}_{j \in [W]}$. Consequently, if ALG successfully solves the *strong $\varepsilon$* exploration problem on $S$, then the set $\{\widehat{\mathrm{arm}}_i\}_{i \in [\frac{n}{2W}]}$ must be the $\varepsilon$-best arm for the arms $\{\mathrm{arm}_{i,j}\}_{i \in [\frac{n}{2W}], j \in [W]}$. Since ALG accomplishes the *strong $\varepsilon$* exploration task on $S$ with a probability of at least $99/100$, it follows that ALG$'$ solves the pure exploration problem on the arms $\{\mathrm{arm}_{i,j}\}_{i \in [\frac{n}{2W}], j \in [W]}$ with the same probability.

Furthermore, since $\mathrm{arm}_0$ is merely a virtual arm, the actual number of pulls by ALG$'$ is equivalent to the pulls used on the real arms $\{\mathrm{arm}_{i,j}\}_{i \in [\frac{n}{2W}], j \in [W]}$. Therefore, the number of pulls made by ALG$'$ is at most equal to the number of pulls made by ALG when solving the *strong $\varepsilon$* exploration problem on $S$. Hence, we can conclude that $m' \leqslant m$. □

The following is a technical lemma that states a "direct sum" type of bound for solving $k$ independent copies of the same problem.

**Lemma C.6.** *Let $f$ be a function to compute, and let $\mathcal{H}$ be a distribution from which the inputs of $f$ are sampled. Suppose that solving $f$ over the distribution $\mathcal{H}$ with probability $1 - \delta$ takes $\Omega(q \cdot \log(\frac{1}{\delta}))$ queries on*

---

**Algorithm 2:** ALG′: Algorithm Transformation

---

**Input:** Arms $\{\text{arm}_{i,j}\}_{i\in[\frac{n}{2W}],j\in[W]}$, algorithm ALG;
**Output:** A set of arms $\{\widehat{\text{arm}}_i\}_{i\in[\frac{n}{2W}]}$, where $\widehat{\text{arm}}_i$ is an $\varepsilon$-best arm of $\{\widehat{\text{arm}}_{i,j}\}_{j\in[W]}$;
Let $\text{arm}_0$ be the arm always return 0;
**for** $i \leftarrow 1$ *to* $\frac{n}{2W}$ **do**
  **for** $j \leftarrow 1$ *to* $W$ **do**
    $\text{arm}'_{(i-1)\cdot 2W+j} \leftarrow \text{arm}_0$;
    $\text{arm}'_{i\cdot 2W+j} \leftarrow \text{arm}_{i,j}$;
  **end**
**end**
Build stream $S = \{\text{arm}'_k\}_{k=1}^n$;
$\{\widetilde{\text{arm}}_k\}_{k=1}^n \leftarrow \text{ALG}(S)$;
**for** $i \leftarrow 1$ *to* $\frac{n}{2W}$ **do**
  $\widehat{\text{arm}}_i \leftarrow \widetilde{\text{arm}}_{i\cdot 2W}$;
**end**
**return** $\{\widehat{\text{arm}}_i\}_{i\in[\frac{n}{2W}]}$

---

*the input. Furthermore, let $\widetilde{\mathcal{H}} = (\mathcal{H}_1, \mathcal{H}_2, \cdots, \mathcal{H}_k)$ be a distribution over $k$ independent copies of $\mathcal{H}$. Then, any algorithm ALG that computes $f$ on* all *copies with probability at least $99/100$ has to make $\Omega(k \cdot q \cdot \log k)$ total queries.*

*Proof.* The lemma follows from a direct calculation of the success probability, and we provide the proof for the purpose of completeness. Define $\mathcal{E}_i, i \in [k]$ as the event that ALG successfully computes $f$ on the $i$-th copy of $\mathcal{H}$, and define $\mathcal{E}$ as the event that *all* copies of $f$ are correctly computed. We have that

$$\Pr(\mathcal{E}) = \Pr\left(\cap_{i=1}^k \mathcal{E}_i\right)$$

$$= \prod_{i=1}^k \Pr\left(\mathcal{E}_i \mid \cap_{j=1}^{i-1} \mathcal{E}_j\right) \qquad \text{(by the law of total probability)}$$

$$= \prod_{i=1}^k \Pr(\mathcal{E}_i). \qquad \text{(by the independence)}$$

Therefore, by using the condition that success probability is at least $99/100$, we have that

$$\sum_{i=1}^k \log(\Pr(\mathcal{E}_i)) = \log(\Pr(\mathcal{E})) \geqslant \sigma - 1$$

for some $\sigma \in (0.9, 1)$. We claim that for each least $k/100$ indices of $i \in [k]$, there must be $\log(\Pr(\mathcal{E}_i)) \geqslant \log(1 - \frac{\sigma}{k})$. Otherwise, the total success probability is at most

$$\frac{99k}{100} \cdot \log(1 - \frac{\sigma}{k}) + \frac{k}{100} \cdot \log(1) = \frac{99k}{100} \cdot \left(\frac{\ln(1 - \frac{\sigma}{k})}{\ln 2}\right)$$

$$\leqslant \frac{99k}{100 \ln 2} \cdot \left(-\frac{\sigma}{k}\right) \qquad \text{(using } \ln(1 + x) \leqslant x\text{)}$$

$$= -\frac{99\sigma}{100 \ln 2} < -1.28 < \sigma - 1. \qquad \text{(by } \sigma > 0.9\text{)}$$

Since solving each $f$ with probability $1 - \delta$ requires $\Omega(q \cdot \log(\frac{1}{\delta}))$ queries, solving $k/100$ indices with probability at least $1 - \sigma/k$ requires

$$\frac{k}{100} \cdot q \cdot \log(\frac{k}{\sigma}) = \Omega(k \cdot q \cdot \log k)$$

queries, which is as desired. $\qquad\square$

***Finalizing the proof of Lemma 4.1.*** Consider any algorithm ALG that successfully solves the *strong $\varepsilon$* exploration problem with a probability of at least $99/100$ on $\mathcal{D}(n, W, \varepsilon)$. Let $T_{\text{ALG}}$ represent the number of pulls executed by this algorithm. We will define ALG$'$ as the algorithm derived from ALG using Algorithm 2, and let $T_{\text{ALG}'}$ be the corresponding number of pulls for ALG$'$.

According to Lemma C.5, ALG$'$ is capable of solving $\frac{n}{2W}$ independent $\varepsilon$ exploration tasks involving $W$ arms simultaneously, and it holds that $\mathbb{E}\left[T_{\text{ALG}'}\right] \leqslant \mathbb{E}\left[T_{\text{ALG}}\right]$.

Since it is necessary to make $\Omega\left(\frac{W}{\varepsilon^2} \cdot \log\left(\frac{1}{\delta}\right)\right)$ pulls to solve the $\varepsilon$ exploration of $W$ arms with a probability of at least $1 - \delta$, we can employ Lemma C.6 to conclude that $\Omega\left(\frac{n}{\varepsilon^2} \log \frac{n}{W}\right)$ samples are required for the algorithm ALG. $\qquad\square$

## C.3   THE ANALYSIS FOR LEMMA 4.2

The algorithm is still Algorithm 1. We employ a larger pulling size of $l = \frac{9}{2\varepsilon^2} \ln \frac{6n}{\delta}$ compared to the weak exploration. This adjustment allows us to effectively apply a union bound across $n$ arms. The increased pulling size not only ensures that we can accurately identify all the $\epsilon$-best arms simultaneously with high probability, but it also results in higher sample complexity.

The following claim establishes bounds on both the *space complexity* and *sample complexity* of this algorithm.

**Lemma C.7.** *The space complexity of the* BUCKET $\left(\frac{9}{2\varepsilon^2} \ln \frac{6n}{\delta}\right)$ *is* $O\left(\frac{1}{\varepsilon}\right)$, *and the sample complexity is* $O\left(\frac{n}{\varepsilon^2} \log \frac{n}{\delta}\right)$.

The proof follows the same reasoning as the proofs of Lemma C.1 and Lemma C.2, and we skip the details to avoid repetitions. Next, we will demonstrate the correctness of the algorithm. We use the notation $\mu(\texttt{arm})$ to denote the mean of the arm.

**Lemma C.8.** *Let* $A = \{\widehat{arm}_t\}_{t=1}^n$ *be the set of arms outputted by the algorithm, and let* $A' = \{arm^*(W, t)\}_{t=1}^n$ *represent the set of best arms. Then, with a probability of at least* $1 - \delta$, *it holds that* $\mu(\widehat{arm}_t) \geqslant \mu(arm^*(W, t)) - \varepsilon$ *for all time* $t \in [n]$.

*Proof.* Let $b_i$ denote the index of the correct bucket to which arm $\texttt{arm}_i$ belongs. In other words, we have $\mu_i \in \left((b_i - 1)\frac{\varepsilon}{3}, b_i \frac{\varepsilon}{3}\right]$. We define $b_i'$ as the index of the bucket where $\texttt{arm}_i$ is stored upon its arrival, which implies $\widehat{\mu}_i \in \left((b_i' - 1)\frac{\varepsilon}{3}, b_i' \frac{\varepsilon}{3}\right]$.

Let $\mathcal{E}_i$ be the event that $|b_i - b_i'| \leqslant 1$, indicating that $\texttt{arm}_i$ is stored in a bucket close to its correct bucket. According to Lemma A.2, we have:

$$\mathbf{Pr}\left[\neg\mathcal{E}_i\right] = \mathbf{Pr}\left[|b_i - b_i'| > 1\right] \leqslant \mathbf{Pr}\left[|\widehat{\mu}_i - \mu_i| > \frac{\varepsilon}{3}\right] \leqslant 2 \cdot \exp\left(-2l \cdot \frac{\varepsilon^2}{9}\right) = \frac{\delta}{3n}.$$

By applying the union bound, we can express this as:

$$\mathbf{Pr}\left[\cap_{i=1}^n \mathcal{E}_i\right] = 1 - \mathbf{Pr}\left[\neg \cap_{i=1}^n \mathcal{E}_i\right] = 1 - \mathbf{Pr}\left[\cup_{i=1}^n \neg\mathcal{E}_i\right] \qquad \text{(By De Morgan's Law)}$$

$$\geqslant 1 - \sum_{i=1}^n \mathbf{Pr}\left[\neg\mathcal{E}_i\right] \geqslant 1 - \sum_{i=1}^n \frac{\delta}{3n} = 1 - \frac{\delta}{3}.$$

If the event $\cap_{i=1}^{n} \mathcal{E}_i$ occurs, it means each arm is placed in a bucket $b_i'$ that is close to its correct bucket $b_i$, satisfying $|b_i - b_i'| \leqslant 1$.

For any $t \in [n]$, suppose that $\widehat{\mathrm{arm}}_t = \mathrm{arm}_{i_t}$ and $\mathrm{arm}^*(W, t) = \mathrm{arm}_{j_t}$. Therefore, we have $|b_{i_t} - b_{i_t}'| \leqslant 1$ and $|b_{j_t} - b_{j_t}'| \leqslant 1$. Additionally, since $\mathrm{arm}^*(W, t) = \mathrm{arm}_{j_t}$ does not expire at time $t$, it follows that the bucket $B_{j_t} \neq \emptyset$ at time $t$. Given that the arm returned by the algorithm at time $t$ is $\widehat{\mathrm{arm}}_t = \mathrm{arm}_{i_t}$, it must be that $b_{i_t}' \geqslant b_{j_t}'$. Thus, we can derive:

$$b_{i_t} \geqslant b_{i_t}' - 1 \geqslant b_{j_t}' - 1 \geqslant b_{j_t} - 2.$$

Consequently, we obtain $\mu(\widehat{\mathrm{arm}}_t) \geqslant \mu(\mathrm{arm}^*(W, t)) - \frac{2}{3}\varepsilon$. $\qquad\square$

## D   THE COMPLETE DETAILS FOR RESULTS IN SECTION 5

We provide the proof of Theorem 3 in this section.

*Proof.* We proceed with the lower bound proof first. According to Yao's minimax principle Yao (1977), it is sufficient to establish the lower bound for deterministic algorithms under a challenging distribution of inputs. Consider the following distribution of $n$ arms.

---

**EPOCH$(n, W)$: A hard distribution with $n$ arms for epoch-wise regret minimization**

1. For $i = k \cdot 2W + j$, where $j \in [W]$, $\mu_i = 1 - \frac{j}{W}$.

2. With probability $\frac{1}{W}$, choose $h \in [W]$ uniformly. For $i = k \cdot 2W + W + j$, where $j \in [W]$, $\mu_i = 1 - \frac{2h+1}{2W}$.

---

Since we have only $\frac{W-1}{2}$ space available, there exists an arm $\mathrm{arm}_i$ where $i \in [k \cdot 2W + 1, k \cdot 2W + W]$ that we cannot store at time $k \cdot 2W + W$. Let $\mathcal{E}_i$ be the event where the $W$ subsequent arms all have the same mean of $1 - \frac{2h+1}{2W}$, with $h \equiv i \pmod{2W}$, but $\mathrm{arm}_i$ is not stored at time $k \cdot 2W + W$. When event $\mathcal{E}_k$ occurs, $\mathrm{arm}_i$ is the best arm at time $i + W - 1$. As we missed $\mathrm{arm}_i$, this will induce at least $\frac{1}{2W} \cdot \frac{T}{n - W + 1}$ regret during this epoch.

Let

$$\mathcal{F}_k = \bigcup_{i=k \cdot 2W + 1}^{k \cdot 2W + W} \mathcal{E}_i.$$

The event $\mathcal{F}_k$ represents that at least one of the events $\mathcal{E}_i$ occurs between the times $[k \cdot 2W + 1, (k+1) \cdot 2W]$. Since at least half of the arms among $\{\mathrm{arm}_{k \cdot 2W + 1}, \cdots, \mathrm{arm}_{k \cdot 2W + W}\}$ are not stored at time $k \cdot 2W + W$, we have $\mathbf{Pr}[\mathcal{F}_k] \geqslant \frac{1}{2}$.

Let $X_k$ be the random variable indicating whether $\mathcal{F}_k$ occurs. The total number of such events that occur is $Y = \sum_{k=1}^{m} X_k$, where $m = \lfloor \frac{n}{2W} \rfloor$. Since $X_i$ are independent Bernoulli random variables with probability $p \geqslant \frac{1}{2}$, $Y$ follows a binomial distribution $Y \sim \mathrm{Bino}(m, p)$. Let $Z \sim \mathrm{Bino}(m, \frac{1}{2})$.

We can analyze the probability as follows:

$$\mathbf{Pr}\left[Y \leqslant \frac{m}{4}\right] \leqslant \mathbf{Pr}\left[Z \leqslant \frac{m}{4}\right] \qquad\qquad \text{(since } p \geqslant \tfrac{1}{2}\text{)}$$

$$\leqslant \mathbf{Pr}\left[\left|Z - \frac{m}{2}\right| \geqslant \frac{m}{4}\right] \leqslant \frac{4}{m} \qquad\qquad \text{(by Chebyshev's inequality)}$$

$$\leqslant \frac{1}{2}. \qquad\qquad \text{(because } m = \lfloor \tfrac{n}{2W} \rfloor \text{ and } n \geqslant 16W\text{)}$$

Let $\mathcal{G}$ be the event that $Y \geqslant \frac{m}{4}$. Then we have:

$$\mathbb{E}\left[R_T\right] = \mathbb{E}\left[R_T \mid \mathcal{G}\right] \cdot \mathbf{Pr}\left[\mathcal{G}\right] + \mathbb{E}\left[R_T \mid \neg\mathcal{G}\right] \cdot \mathbf{Pr}\left[\neg\mathcal{G}\right] \geqslant \mathbb{E}\left[R_T \mid \mathcal{G}\right] \cdot \frac{1}{2}.$$

Since at least $\frac{m}{4}$ of the events $\mathcal{F}_k$ occur when $\mathcal{G}$ occurs, and each event $\mathcal{F}_k$ induces at least $\frac{1}{2W} \cdot \frac{T}{n-W+1}$ regret, we have:

$$\mathbb{E}\left[R_T \mid \mathcal{G}\right] \geqslant \frac{m}{4} \cdot \frac{1}{2W} \cdot \frac{T}{n-W+1}.$$

Hence, we find:

$$\mathbb{E}\left[R_T\right] \geqslant \frac{m}{4} \cdot \frac{1}{2W} \cdot \frac{T}{n-W+1} \cdot \frac{1}{2} \geqslant \frac{m \cdot T}{16 \cdot W \cdot n}$$

$$\geqslant \frac{n}{4W} \cdot \frac{T}{16 \cdot W \cdot n} \qquad\qquad \text{(because } m = \lfloor \tfrac{n}{2W} \rfloor \geqslant \tfrac{n}{4W}\text{)}$$

$$= \frac{T}{64 \cdot W^2}.$$

For the upper bound, we proceed by running epoch-wise UCB using the $W$ memory size. There are algorithms, such as INF Audibert & Bubeck (2009), that can achieve a total regret of $O(\sqrt{nT})$ in a centralized setting with $n$ arms. We can utilize such an algorithm as a black box to attain a total regret of $O(\sqrt{W \cdot (n-W) \cdot T})$. The strategy is straightforward: we apply the INF algorithm to each epoch of the stream. Since we need to pull $\frac{T}{n-W+1}$ times for each epoch, this approach will result in a total regret of $O(\sqrt{WT/(n-W)})$ for each epoch. Consequently, the overall total regret will be $O(\sqrt{W \cdot (n-W) \cdot T})$. Theorem 3 $\square$

**Remark 2.** Note that the proof in this section naturally generalizes to the setting with *non-uniform* number of samples per epoch. In particular, on the lower bound side, we can uniformly at random sample one window $i^* \in [k \cdot 2W + 1, k \cdot 2W + W]$ and allocate all the $T$ samples in the window. With probability at least $1/2$, the algorithm cannot store the arm with mean more than $1 - \frac{2h+1}{2W}$, which means the induced regret is again at least $\Omega(T/W)$ in this epoch. For the upper bound with $\Omega(W)$-arm memory, if we let $T_k$ be the number of trials in epoch $k$, we can obtain $O(\sum_{k=1}^{n-W+1} \sqrt{WT_k})$ regret, although the regret bound is less informative.

# E  REGRET MINIMIZATION WITH AN EVERLASTING BEST ARM

Our main regret notion is based on the *epoch-wise* regret. However, in some practical scenarios, there are cases where the most popular item is not limited by the time horizon. Consider, again, the task of recommending movies to users for entertainment companies. On average, a movie remains in theaters for about 1 to 2 months. However, some exceptionally popular pieces can have a much longer run. For example, *The Sound*

*of Music* was screened in theaters for 4 years and 6 months, while *Avatar* stayed for 34 weeks. Therefore, when designing a recommendation system for currently showing movies, we can assume a sliding window of 2 months, but there are some enduring ones that remain popular even after this window has passed.

Similar situations occur in other contexts as well. For instance, the song *Lose Control* set a new record by spending 107 weeks on the Billboard Hot 100 chart, whereas the average lifespan of a song on the chart is typically between 6 and 7 weeks.

Inspired by these applications, we also propose another regret model, which we refer to as **regret minimization with an everlasting best arm**. In the scenario where there is an everlasting best arm, we can pull this best arm even if it appears more than $W$ time units earlier in the stream. For situations involving such an everlasting best arm, there are two slightly different scenarios to consider: whether we are allowed to pull a sub-optimal expired arm and get 1 regret, or we could simply get a signal that a sub-optimal arm is expired and the sampling operation is disallowed.

### E.1 REGRET MINIMIZATION WITH EVERLASTING BEST ARM AND EXPLICIT VALID FLAG

The first scenario is when we cannot pull an expired arm other than the best arm. This means that the only arms available for pulling are the everlasting best arm and the $W$ most recent arms, and all expired arms (other than the best) in the memory will carry a flag of "being invalid". A practical example of this scenario might be recommending currently showing movies. If a movie is still being presented by the theater after the sliding-window period, it indicates that the movie has not expired and can still be selected. Therefore, in this setting, we can assume the existence of a flag for each arm indicating whether it is valid for pulling. Only the everlasting best arm and the $W$ most recent arms will have a positive flag.

**Definition 7** (Valid flag). Let $\{\texttt{arm}_i\}_{i=1}^n$ be a collection of $n$ arms with an everlasting best arm denoted as $\texttt{arm}^*$, and let $W$ be the window size. The valid flag is a function $\texttt{flag}(\texttt{arm}_i, t)$ that returns $\texttt{True}$ if $\texttt{arm}_i$ is $\texttt{arm}^*$ or if $i \geqslant t - W + 1$ (indicating that $\texttt{arm}_i$ is one of the $W$ most recent arms); it returns $\texttt{False}$ otherwise.

In simpler terms, $\texttt{flag}(\texttt{arm}_i, t) = \texttt{True}$ if and only if $\texttt{arm}_i$ is valid at time $t$.

Next, we define regret minimization with an everlasting best arm and an explicit valid flag. In this setting, the $\texttt{flag}$ function is accessible to the algorithm, allowing it to determine whether an arm is valid without needing to pull it. This aligns with scenarios such as recommending movies that are currently showing, where we can ascertain whether a movie is still valid (i.e., still being presented in the theater) without any action.

**Definition 8** (Regret minimization with everlasting best arm and explicit valid flag). Let $\{\texttt{arm}_i\}_{i=1}^n$ represent a collection of $n$ arms with an everlasting best arm, $\texttt{arm}^*$. Let $W$ be the window size and $T$ be the total number of trials. Denote $\mu^*$ as the mean reward of the best arm $\texttt{arm}^*$ (among all arms), and let $t$ denote the variable for the index of the arriving arm. In this scenario, there exists an explicit $\texttt{flag}$ function, and an arm $\texttt{arm}_i$ can be pulled at time $t$ only if $\texttt{flag}(\texttt{arm}_i, t) = \texttt{True}$. Let $\{i(\tau)\}_{\tau=1}^T$ be the set of indices of arms pulled by some algorithm, and the regret is defined as $R_T := \sum_{\tau=1}^T (\mu^* - \mu_{i(\tau)})$.

If we have $W$ memory, a straightforward algorithm is to *not* pull any arm until $W$ steps and check whether the arm is still valid. The arm is valid if and only if it is the best arm, and we could therefore commit to the arm to achieve 0 regret. [6] As such, a natural question is whether we could do better with $o(W)$ memory. In

---

[6]In this scenario, we assume that time continues to pass even when there are no more input arms available. Therefore, every arm, except for the everlasting best arm— including arm $\texttt{arm}_n$—may eventually expire as time goes on. Thus, we can simply wait long enough and use the $\texttt{flag}$ function to identify the everlasting best arm, which will be the only valid arm remaining.

An alternative assumption is that time ceases to progress when there are no new arms in the stream. In this case, the final valid arms will consist of the everlasting best arm plus the last $W$ arms, which are $\{\texttt{arm}_{n-W+1}, \cdots, \texttt{arm}_n\}$. If the

what follows, we will show that the answer to the above question is *negative*: we will show that $\Omega(W)$ space is necessary to achieve a total regret of $o(T)$, essentially indicating that the $W$-memory algorithm is *optimal*.

**Theorem 4.** *For any given parameters $T$, $n$, and $W$ such that $T \geqslant n \geqslant 4W$, there exists a family of streaming stochastic multi-armed bandit instances such that any single-pass streaming algorithm designed for a sliding-window stream of length $n$ with a window size $W$ and a memory of $\frac{W}{8}$ arms must incur a total expected regret of at least*

$$\mathbb{E}\left[R_T\right] \geqslant \frac{T}{120}.$$

*Furthermore, there exists an algorithm that given a stream of $n$ arms and parameters $W$ and $T$, achieves $0$ regret with a memory of $W$ arms.*

Theorem 4 shows an extremely sharp "phase transition" for the memory-regret trade-off: with $o(W)$ memory, we have to suffer $\Omega(T)$ regret. On the other hand, if we slightly increase the memory to $W$, we could achieve $0$ total regret.

To prove Theorem 4 we will utilize the following result on the sample-memory trade-offs for *storing an arm* in the memory from Assadi & Wang (2022).

**Proposition E.1** (Assadi & Wang (2022), cf. Chen et al. (2024)). *Consider the following distribution of $m$ arms.*

---

**DIST**$(m, \sigma, \beta)$*: A hard distribution with $m$ arms for trapping the best arm*

*1. An index $i^*$ sampled uniform at random from $[m]$.*

*2. For $i \neq i^*$, let the arms be with reward $\mu_i = \sigma$.*

*3. For $i = i^*$, let the arm be with reward $\mu_{i^*} = \sigma + \beta$.*

---

*Any algorithm that outputs (the indices of) $\frac{m}{8}$ arms that contain the best arm on DIST with a probability of at least $\frac{2}{3}$ has to use at least $\frac{1}{1200} \cdot \frac{m}{\beta^2}$ arm pulls.*

The intuition behind our proof is that it is crucial not to overlook the best arm in a stream of options. By utilizing the distribution DIST, we can construct scenarios where it requires a considerable number of pulls to identify the best arm. Additionally, we can create instances that include multiple distributions of DIST, making it challenging to determine the best arm.

In these scenarios, an algorithm that uses a large number of arm pulls on earlier arms risks the possibility that the best arm may appear later in the stream. If the algorithm has already made too many pulls on suboptimal arms, it will incur substantial regret. Conversely, if the algorithm decides to conserve its pulls and primarily engages with the later part of the stream, there is a risk that the best arm could arrive early on, leading the algorithm to miss it entirely. As a result, any algorithm faces the inherent risk of missing the best arm, which can lead to significant regret.

***The proof of Theorem 4.*** In the discussion by the start of Section E.1, we have already introduced the relatively simple algorithm that uses $W$ arm memory and achieves $0$ regret. We focus on proving the lower bound in the proof.

---

everlasting best arm is not included among the last $W$ arms, we can identify it directly. If it is among the last $W$ arms, the problem then shifts to a centralized regret minimization problem with those $W$ arms. This represents a combination of our initial setting and the centralized regret minimization framework, so we will forego further discussion of this assumption.

According to Yao's minimax principle Yao (1977), it is sufficient to establish the lower bound for deterministic algorithms over a challenging distribution of inputs.

We will first introduce the CONST distribution for clarity.

---

**CONST$(m, \sigma)$: A distribution with $m$ arms with the same means**

1. $\forall i \in [m], \mu_i = \sigma$.

---

Let $\beta = \min\{\sqrt{\frac{W}{1200T}}, \frac{1}{10}\}$. Consider the following distribution of $n$ arms.

---

**SIGNAL$(n, W, \beta)$: A hard distribution with $n$ arms for regret minimization with an everlasting best arm with expiation signal**

1. The first $W$ arms of SIGNAL$(n, W, \beta)$ is DIST$(W, \frac{3}{5}, \beta)$.

2. The $W + 1$-th to $2W$-th arms are CONST$(W, \frac{1}{5})$.

3. With probability of $\frac{1}{2}$, $2W + 1$-th to $3W$-th arms are DIST$(W, \frac{2}{5}, \beta)$, otherwise, DIST$(W, \frac{4}{5}, \beta)$.

4. The remaining $n - 3W$ arms are CONST$(n - 3W, \frac{1}{5})$.

---

For any deterministic algorithm ALG, let $T_{\text{ALG}}$ denote the number of pulls it uses until time $2W$. Since ALG is a deterministic algorithm and the first $2W$ arms are the same in both scenarios, $T_{\text{ALG}}$ remains consistent regardless of whether the arms from $2W + 1$ to $3W$ follow the distribution DIST$(W, \frac{2}{5}, \beta)$ or DIST$(W, \frac{4}{5}, \beta)$.

Let $\mathcal{E}_1$ be the event that the arms from $2W + 1$ to $3W$ are DIST$(W, \frac{2}{5}, \beta)$ and $\mathcal{E}_2$ be the event that they are DIST$(W, \frac{4}{5}, \beta)$. If $T_{\text{ALG}} \leqslant \frac{T}{2}$, then we have:

$$\mathbb{E}[R_T] = \mathbb{E}[R_T|\mathcal{E}_1] \cdot \mathbf{Pr}[\mathcal{E}_1] + \mathbb{E}[R_T|\mathcal{E}_2] \cdot \mathbf{Pr}[\mathcal{E}_2] \geqslant \mathbb{E}[R_T|\mathcal{E}_1] \cdot \mathbf{Pr}[\mathcal{E}_1].$$

Since $\frac{1}{1200} \cdot \frac{W}{\beta^2} \geqslant T > \frac{1}{2}T \geqslant T_{\text{ALG}}$, by Lemma E.1, the probability that the best arm is stored in memory at time $W + 1$ is at most $\frac{2}{3}$. Let $\mathcal{F}$ be the event that the best arm is stored in memory at time $W + 1$. Then,

$$\mathbb{E}[R_T \mid \mathcal{E}_1] = \mathbb{E}[R_T \mid \mathcal{E}_1 \cap \mathcal{F}] \cdot \mathbf{Pr}[\mathcal{F}] + \mathbb{E}[R_T \mid \mathcal{E}_1 \cap \neg\mathcal{F}] \cdot \mathbf{Pr}[\neg\mathcal{F}] \geqslant \mathbb{E}[R_T \mid \mathcal{E}_1 \cap \neg\mathcal{F}] \cdot \mathbf{Pr}[\neg\mathcal{F}].$$

When the event $\mathcal{E}_1 \cap \neg\mathcal{F}$ occurs, the best arm is not stored in memory at time $2W + 1$, and all the arms with means $\frac{3}{5}$ have expired. The remaining arms have means at most $\frac{2}{5} + \beta \leqslant \frac{2}{5} + \frac{1}{10} = \frac{1}{2}$. Since we can only pull valid arms, the arms we can pull have a mean reward of at most $\frac{1}{2}$. Thus, the regret for each pull after time $2W$ is at least

$$\frac{3}{5} + \beta - \frac{1}{2} \geqslant \frac{1}{10}.$$

Therefore, we have:

$$\mathbb{E}[R_T \mid \mathcal{E}_1 \cap \neg\mathcal{F}] \geqslant (T - T_{\text{ALG}}) \cdot \frac{1}{10} \geqslant \left(T - \frac{T}{2}\right) \cdot \frac{1}{10} = \frac{T}{20}.$$

Thus,

$$\mathbb{E}[R_T] \geqslant \mathbb{E}[R_T \mid \mathcal{E}_1] \cdot \mathbf{Pr}[\mathcal{E}_1] \geqslant \mathbb{E}[R_T \mid \mathcal{E}_1 \cap \neg\mathcal{F}] \cdot \mathbf{Pr}[\neg\mathcal{F}] \cdot \mathbf{Pr}[\mathcal{E}_1] \geqslant \frac{T}{20} \cdot \frac{1}{3} \cdot \frac{1}{2} = \frac{T}{120}.$$

On the other hand, if $T_{\text{ALG}} \geqslant \frac{T}{2}$, then we have:

$$\mathbb{E}\left[R_T\right] \geqslant \mathbb{E}\left[R_T \mid \mathcal{E}_2\right] \cdot \mathbf{Pr}\left[\mathcal{E}_2\right].$$

When event $\mathcal{E}_2$ occurs, the best arm has a mean of $\frac{4}{5} + \beta$. The regret for each pull on the first $2W$ arms would be at least

$$\frac{4}{5} + \beta - \frac{3}{5} - \beta = \frac{1}{5}.$$

Therefore,

$$\mathbb{E}\left[R_T\right] \geqslant \frac{1}{2} \cdot \mathbb{E}\left[R_T \mid \mathcal{E}_2\right] \geqslant \frac{1}{2}T_{\text{ALG}} \cdot \frac{1}{5} \geqslant \frac{T}{20}.$$

In conclusion, since $\mathbb{E}\left[R_T\right] \geqslant \frac{T}{120}$ regardless of whether $T_{\text{ALG}} \geqslant \frac{T}{2}$ or not, we have completed our proof.

Theorem 4 $\square$

## E.2 REGRET MINIMIZATION WITH EVERLASTING BEST ARM AND IMPLICIT VALID FLAG

The second scenario is when we can still pull an expired arm, but doing so incurs a significant penalty. In this situation, any arm can be pulled at any time; however, if an expired arm is selected, a regret penalty of 1 is incurred.

Furthermore, we assume that the penalty associated with pulling an expired arm is not immediately known. If we were to be instantly informed about any penalties, the scenario could be simplified: we would incur a 1 regret penalty for all non-best arms in an effort to identify the best arm. This would lead to a total regret of $n - 1$, which is negligible given that $T \gg n$.

In this context, we can still assume the existence of a valid flag function, denoted as flag, to indicate whether an arm is valid. However, the algorithm cannot access this flag; therefore, it cannot determine whether an arm is valid.

**Definition 9** (Regret minimization with an everlasting best arm and implicit valid flag). Let $\{\text{arm}_i\}_{i=1}^n$ represent a collection of $n$ arms with an everlasting best arm, $\text{arm}^*$. Let $W$ be the window size and $T$ be the total number of trials. Denote $\mu^*$ as the mean reward of the best arm $\text{arm}^*$ (among all arms), and let $t$ denote the variable for the index of the arriving arm. In this scenario, an implicit flag function exists. Any arm $\text{arm}_i$ can be pulled at any time $t$, but if the flag indicates it is invalid (i.e., $\text{flag}(\text{arm}_i, t) = \text{False}$), a regret of 1 is incurred. Let $\{i(\tau)\}_{\tau=1}^T$ be the set of indices of arms pulled by a given algorithm, and let $\{\text{flag}_\tau = \text{flag}(\text{arm}_{i(\tau)}, t)\}_{\tau=1}^T$ represent the validity flag for the arms that were pulled. The total regret is defined as $R_T := \sum_{\text{flag}_\tau = \text{True}}(\mu^* - \mu_{i(\tau)}) + \sum_{\text{flag}_\tau = \text{False}} 1$.

In this setting, an $\Omega(W)$ memory is still necessary to achieve a total regret of $o(T)$. Although there is an everlasting best arm, the lack of a signal about whether an arm has expired makes this setting strictly more challenging than the one in which we receive an expiry signal. Thus, the claims and proofs applicable to the setting with signals also remain valid in this case.

**Lemma E.2.** *There exists a family of streaming stochastic multi-armed bandit instances such that, for any given parameters $T$, $n$, and $W$, where $T \geqslant n \geqslant 4W$, any single-pass streaming algorithm for a sliding-window stream of length $n$ with a window size $W$ and a memory of $\frac{W}{8}$ arms must incur a total expected regret of at least*

$$\mathbb{E}\left[R_T\right] \geqslant \frac{T}{120}.$$

However, in this setting, the upper bound of total regret with trivial space complexity $W - 1$ is no longer $O_{n,W}(1)$. Since we are not informed whether an arm is expired, it becomes challenging to identify the best

arm easily. When arms have means close to the best arm, we must pull these arms numerous times, which leads to significant regret, as each pull on an expired arm incurs a regret penalty of 1. Furthermore, even after many pulls on these arms, we may still incorrectly identify the best arm, resulting in additional regret. In fact, achieving $o(T)$ total regret is impossible even with a memory capacity of $n - 1$, which is an even stronger condition than $W - 1$ memory.

**Lemma E.3.** *There exists a family of streaming stochastic multi-armed bandit instances such that, for any given parameters $T$, $n$, and $W$ with $T \geqslant n \geqslant 8W$, any single-pass streaming algorithm for a sliding window stream of length $n$, with a window size $W$ and a memory of $n - 1$ arms, must incur*

$$\mathbb{E}[R_T] \geqslant \frac{T}{1800}$$

*total expected regret.*

The intuition behind this is that we cannot identify the best arm even after $T$ pulls in $\mathrm{DIST}(W, \frac{3}{5}, \beta)$ with $\beta = O\left(\frac{1}{T}\right)$, since $O\left(\frac{W}{\varepsilon^2}\right)$ pulls are required to identify the $\varepsilon$-best arm among $W$ arms. In this scenario, when the arm is not expired, it will incur $\beta$ regret per pull, while an expired arm incurs a regret of 1. Thus, a sound strategy would be to pull the arms before they expire. However, if there are two such distributions in the stream, we cannot determine in advance which distribution contains the best arm. Consequently, we risk either overspending pulls on the wrong distribution or incurring 1 regret per pull by not allocating most pulls to the correct distribution. This results in a total regret of $\Theta(T)$ in either situation.

*Proof.* By Yao's minimax principle Yao (1977), it is sufficient to demonstrate the lower bound for deterministic algorithms in the face of a challenging distribution of inputs. Let's consider the following distribution:

---

**SIGNAL$'(n, W, \beta)$: A hard distribution with $n$ arms for regret minimization with an everlasting best arm with expiation signal**

1. The first $W$ arms of $\mathrm{SIGNAL}(n, W, \beta)$ is $\mathrm{DIST}(W, \frac{3}{5}, \beta)$.

2. The $W + 1$-th to $2W$-th arms are $\mathrm{CONST}(W, \frac{1}{5})$.

3. With probability of $\frac{1}{2}$, $2W + 1$-th to $3W$-th arms are $\mathrm{DIST}(W, \frac{2}{5}, \beta)$, otherwise, $\mathrm{DIST}(W, \frac{4}{5}, \beta)$.

4. The remaining $n - 3W$ arms are $\mathrm{CONST}(n - 3W, \frac{1}{5})$.

---

According to Lemma C.4, there exist constants $c_1$ and $c_2$ such that any algorithm using no more than $c_1 \frac{W}{\beta^2} \log c_2$ pulls will not reliably return the $\frac{\beta}{2}$-best arm with a probability of at least $\frac{3}{4}$. We set $\beta = \min\{\frac{1}{10}, \frac{c_1 W}{2T} \log c_2\}$.

Now, consider $\mathrm{SIGNAL}'(n, W, \beta)$. Let $T_1$ denote the number of pulls made before time $2W + 1$, $T_2$ the number of pulls made between times $2W + 1$ and $4W$, and $T_3$ the number of pulls made after time $4W$. Let $\mathcal{E}_i$ represent the event that $T_i \geqslant \frac{T}{3}$. Since $T_1 + T_2 + T_3 = T$, at least one of the events $\mathcal{E}_i$ must occur. Thus, we have:

$$\mathbb{E}[R_T] = \mathbb{E}[R_T \mid \mathcal{E}_i] \cdot \mathbf{Pr}[\mathcal{E}_i] + \mathbb{E}[R_T \mid \neg\mathcal{E}_i] \cdot \mathbf{Pr}[\neg\mathcal{E}_i] \geqslant \mathbb{E}[R_T | \mathcal{E}_i] \cdot \mathbf{Pr}[\mathcal{E}_i].$$

Therefore, it suffices to show that $\mathbb{E}[R_T \mid \mathcal{E}_i] \geqslant \frac{T}{600}$ for each $i \in [3]$.

**Case 1: $\mathcal{E}_1$ occurs.** Let $\mathcal{F}_1$ be the event that the arms from $2W + 1$ to $3W$ are drawn from $\text{DIST}(W, \frac{2}{5}, \beta)$, and let $\mathcal{F}_2$ be the event that these arms are drawn from $\text{DIST}(W, \frac{4}{5}, \beta)$. Hence,

$$\mathbb{E}\left[R_T \mid \mathcal{E}_1\right] \geqslant \mathbb{E}\left[R_T \mid \mathcal{E}_1 \cap \mathcal{F}_2\right] \cdot \mathbf{Pr}\left[\mathcal{F}_2\right] = \mathbb{E}\left[R_T \mid \mathcal{E}_1 \cap \mathcal{F}_2\right] \cdot \frac{1}{2}.$$

When $\mathcal{F}_2$ occurs, each pull on the first $2W$ arms results in at least $\frac{1}{10}$ regret. Since we spend at least $\frac{T}{3}$ pulls on these first $2W$ arms when $\mathcal{E}_1$ occurs, we have:

$$\mathbb{E}\left[R_T \mid \mathcal{E}_1\right] \geqslant \mathbb{E}\left[R_T \mid \mathcal{E}_1 \cap \mathcal{F}_2\right] \cdot \frac{1}{2} \geqslant \frac{1}{10} \cdot \frac{T}{3} \cdot \frac{1}{2} = \frac{T}{60}.$$

**Case 2: $\mathcal{E}_2$ occurs.** Similarly,

$$\mathbb{E}\left[R_T \mid \mathcal{E}_2\right] \geqslant \mathbb{E}\left[R_T \mid \mathcal{E}_2 \cap \mathcal{F}_1\right] \cdot \mathbf{Pr}\left[\mathcal{F}_1\right] = \mathbb{E}\left[R_T \mid \mathcal{E}_2 \cap \mathcal{F}_1\right] \cdot \frac{1}{2}.$$

When $\mathcal{F}_1$ occurs, each pull on the arms from $2W + 1$ to $4W$ leads to at least $\frac{1}{10}$ regret. Since we make at least $\frac{T}{3}$ pulls in this range when $\mathcal{E}_2$ occurs,

$$\mathbb{E}\left[R_T \mid \mathcal{E}_2\right] \geqslant \mathbb{E}\left[R_T \mid \mathcal{E}_2 \cap \mathcal{F}_1\right] \cdot \frac{1}{2} \geqslant \frac{1}{10} \cdot \frac{T}{3} \cdot \frac{1}{2} = \frac{T}{60}.$$

**Case 3: $\mathcal{E}_3$ occurs.** Given that $\beta = \frac{c_1 W}{2T} \log c_2$, it is impossible to return the $\frac{\beta}{2}$-best arm (which is the exact best arm in this distribution) with a probability of at least $\frac{3}{4}$ by using a maximum of $T$ pulls. Let $\mathcal{G}$ be the event that we pull at most $\frac{T}{6}$ times on the best arm after time $4W$. If $\mathbf{Pr}\left[\mathcal{G}\right] \leqslant \frac{1}{10}$, we can devise a strategy that distinguishes the best arm from the others, which is impossible. Hence, $\mathbf{Pr}\left[\mathcal{G}\right] \geqslant \frac{1}{10}$.

After time $4W$, pulling a valid arm with a mean of $\frac{1}{5}$ results in at least $\frac{1}{10}$ regret, while pulling from an expired arm incurs a regret of $1$. Thus, we incur at least $\frac{1}{10}$ regret if we do not pull the best arm after time $4W$. When $\mathcal{G}$ occurs, we will spend at least $\frac{T}{6}$ pulls on arms other than the best arm beyond time $4W$, leading to:

$$\mathbb{E}\left[R_T \mid \mathcal{E}_3\right] \geqslant \mathbb{E}\left[R_T \mid \mathcal{E}_3 \cap \mathcal{G}\right] \cdot \mathbf{Pr}\left[\mathcal{G}\right] \geqslant \frac{1}{10} \cdot \frac{T}{6} \cdot \frac{1}{10} = \frac{T}{600}.$$

Since at least one of the events $\mathcal{E}_i$ must occur, we have:

$$\max_{i \in [3]}\{\mathbf{Pr}\left[\mathcal{E}_i\right]\} \geqslant \frac{1}{3}.$$

Therefore,

$$\mathbb{E}\left[R_T\right] \geqslant \max_{i \in [3]}\{\mathbb{E}\left[R_T \mid \mathcal{E}_i\right] \cdot \mathbf{Pr}\left[\mathcal{E}_i\right]\} \geqslant \frac{T}{600} \cdot \max_{i \in [3]}\{\mathbf{Pr}\left[\mathcal{E}_i\right]\} \geqslant \frac{T}{600} \cdot \frac{1}{3} = \frac{T}{1800}.$$

$\square$

# F  FAILURE OF STATE-OF-THE-ART ALGORITHMS IN VANILLA STREAMING MABS

In this section, we will provide simple counterexamples and proofs to illustrate why the state-of-the-art algorithms used in vanilla streaming multi-armed bandits (MABs) do not perform well in sliding-window MABs.

It is important to note that, based on Theorems 1 and 3 that we have established, we have demonstrated that any single-pass streaming algorithm utilizing $o(W)$ memory will fail to address both the weak and strong exploration problems or achieve a regret of $o(T)$. Consequently, the state-of-the-art algorithms from vanilla streaming MABs will also fail in the sliding-window setting, as supported by our theorems. While we have already established this in a general context, we believe that providing a simpler, specific proof related to these algorithms will help readers better understand why algorithms with $o(W)$ memory fail in sliding-window scenarios.

The state-of-the-art algorithm for streaming exploration was proposed by Assadi & Wang (2020). This algorithm can identify the best arm with a probability of at least $1 - \delta$ using only 1 unit of memory and $O\left( \frac{n}{\varepsilon^2} \log\left(\frac{1}{\delta}\right) + \log^2(n) \cdot \frac{\log^2\left(\frac{1}{\delta}\right)}{\varepsilon^3} \right)$ pulls. However, we will present a counterexample demonstrating that the algorithm by Assadi & Wang (2020) cannot effectively address weak exploration in the sliding-window setting, even with a success probability of at least $0.6$.

**Lemma F.1.** *There exists a data stream such that the algorithm by Assadi & Wang (2020) cannot resolve the weak exploration for this stream with a probability of at least $0.6$.*

*Proof.* Consider the data stream defined by $\mu_i = 1 - \frac{i-1}{n}$, which has a decreasing mean. In this scenario, the best arm from time $t = 1$ to $W$ is $\text{arm}_1$, while the best arm at time $t = i$ for $i > W$ is $\text{arm}_{i-W+1}$.

If the algorithm can correctly identify the best arm from time $t = 1$ to $W$ with a probability of at least $0.6$, since it has only 1 unit of memory, the arm stored at that time must be $\text{arm}_1$. Consequently, at time $t = W$, only $\text{arm}_1$ will be stored in memory, causing us to lose access to $\text{arm}_2$ indefinitely, which is the best arm at time $t = W + 1$. Thus, it becomes impossible for the algorithm to return the correct best arm at time $t = W + 1$ with a probability greater than $1 - 0.6 = 0.4$. This indicates that the algorithm fails to resolve weak exploration with a probability of at least $0.6$.

Conversely, if the algorithm fails to return the correct best arm from time $t = 1$ to $W$ with a probability of at least $0.6$, it also indicates a failure in solving weak exploration with a probability of at least $0.6$.

Therefore, it is impossible for the algorithm to successfully resolve weak exploration with a probability of at least $0.6$. □

The state-of-the-art algorithm for regret minimization was proposed by Wang (2023). This algorithm achieves a total regret of $O\left(n^{1/3}T^{2/3}\right)$, which matches the lower bound of regret for streaming multi-armed bandits (MABs), using $\lceil \log^* n \rceil + 1$ units of memory. The key idea behind their algorithm is to first identify an $\varepsilon$-best arm for the entire stream and then allocate the remaining pulls to that $\varepsilon$-best arm. By choosing $\varepsilon = O\left( \sqrt[3]{\frac{n \log n}{T}} \right)$, this approach minimizes the total regret to $O\left(n^{1/3}T^{2/3}\right)$.

However, their algorithm cannot be applied to the sliding-window setting. The strategy of locating an $\varepsilon$-best arm and dedicating all remaining pulls to it is flawed, as the $\varepsilon$-best arm can expire over time, making it unavailable for pulling when it does. Therefore, the algorithm proposed by Wang (2023) is not even well-defined in the sliding-window context.

A natural attempt to adapt the algorithm for the sliding-window setting would be to identify an $\varepsilon$-best arm for each epoch and allocate all remaining pulls for that epoch to the identified arm. However, we will demonstrate that there exist data streams that require the algorithm to utilize at least $O\left(\frac{1}{\varepsilon}\right)$ memory to return the $\varepsilon$-best arm for each epoch.

**Lemma F.2.** *There exist data streams and $\varepsilon \in (0, 1)$ such that it is impossible to return the $\varepsilon$-best arm at any time $t$ with a probability of at least $0.6$ using only $o\left(\frac{1}{\varepsilon}\right)$ memory.*

*Proof.* Let $\varepsilon = \frac{1}{100 \cdot W}$. Define the means of the arms as follows:

$$\mu_i = 1 - 2\varepsilon \cdot \left( i - 1 - 50 \cdot W \cdot \left\lfloor \frac{i}{50 \cdot W} \right\rfloor \right).$$

This means the data stream consists of arms with decreasing means, where each subsequent arm has a mean $2\varepsilon$ smaller than the previous arm. The mean of an arm is reset to $1$ when it reaches $0$.

For this type of data stream, since each subsequent arm has a mean $2\varepsilon$ lower than its predecessor, the $\varepsilon$-best arm at any time $t$ is actually the best arm at that time. Consequently, the algorithm must store the best arm to accurately return the $\varepsilon$-best arm. Given the decreasing nature of the arm means, it is necessary to keep track of all arms in the window, requiring a memory size of $W = \Omega\left(\frac{1}{\varepsilon}\right)$. $\qquad\square$

Thus, since certain data streams exist where $W = o\left(\sqrt[3]{\frac{n \log n}{T}}\right)$, it follows that the adapted algorithm cannot operate with $o(W)$ memory for such streams.

# G  ADDITIONAL SETTINGS AND RESULTS OF THE EXPERIMENTS

We provide additional details and results of the experiments with simulations in different settings (including the regret minimization problem with the everlasting arm regret notion).

## G.1  EXPERIMENTAL RESULTS ON PURE EXPLORATION

We implement Algorithm 1 for the $\varepsilon$ exploration over the streams. Here, if we have $m$ memory, then our input to Algorithm 1 is $\varepsilon = O(1/m)$. We report the *mean, median, and maximum gaps* over the $n - W + 1$ steps for different sizes of memories. The illustration of the results could be shown as in Figures 2 to 4. For the trade-offs between the memory and the number of arm pulls, we only report the $W = 50$ case since it contains the cases for $W \in \{10, 20\}$ up to a very small constant factor.

From the figures, it could be observed that there are generally trade-offs between memory/quality and memory/samples. The trade-off curve for the memory/quality is mostly stable: for the mean and median statistics, the error bar obtained from 10 runs is quite narrow. The sample complexity scales quadratically with the memory, which is consistent with the $1/\varepsilon^2$ term in the sampling rate. Finally, note that the sample complexity does *not* change significantly w.r.t. the number of arms, which is also consistent with the fact that the asymptotic number of arm pulls on each arm is $O(\log W/\varepsilon^2)$, which is independent of $n$.

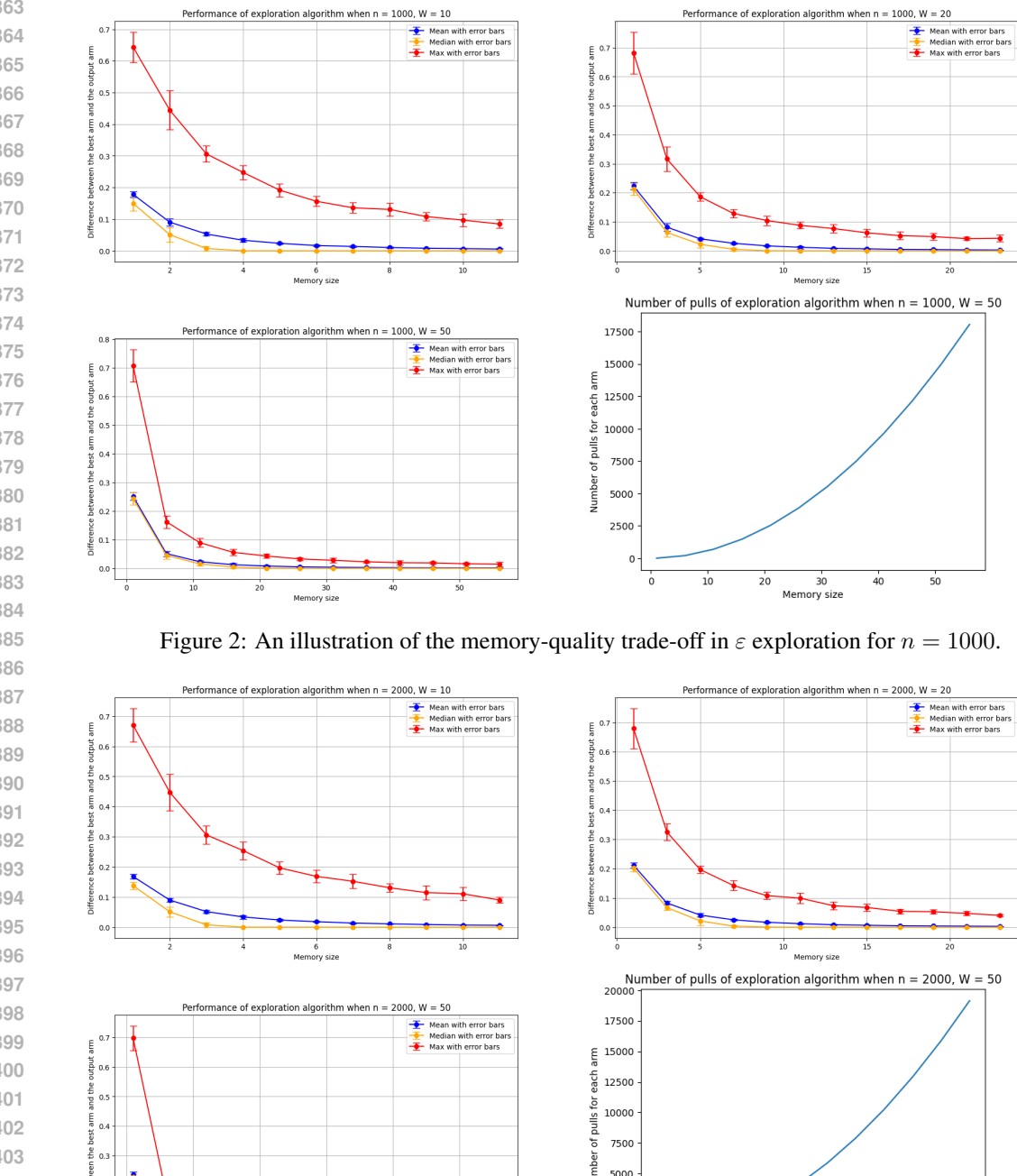

Figure 2: An illustration of the memory-quality trade-off in $\varepsilon$ exploration for $n = 1000$.

Figure 3: An illustration of the memory-quality trade-off in $\varepsilon$ exploration for $n = 2000$.

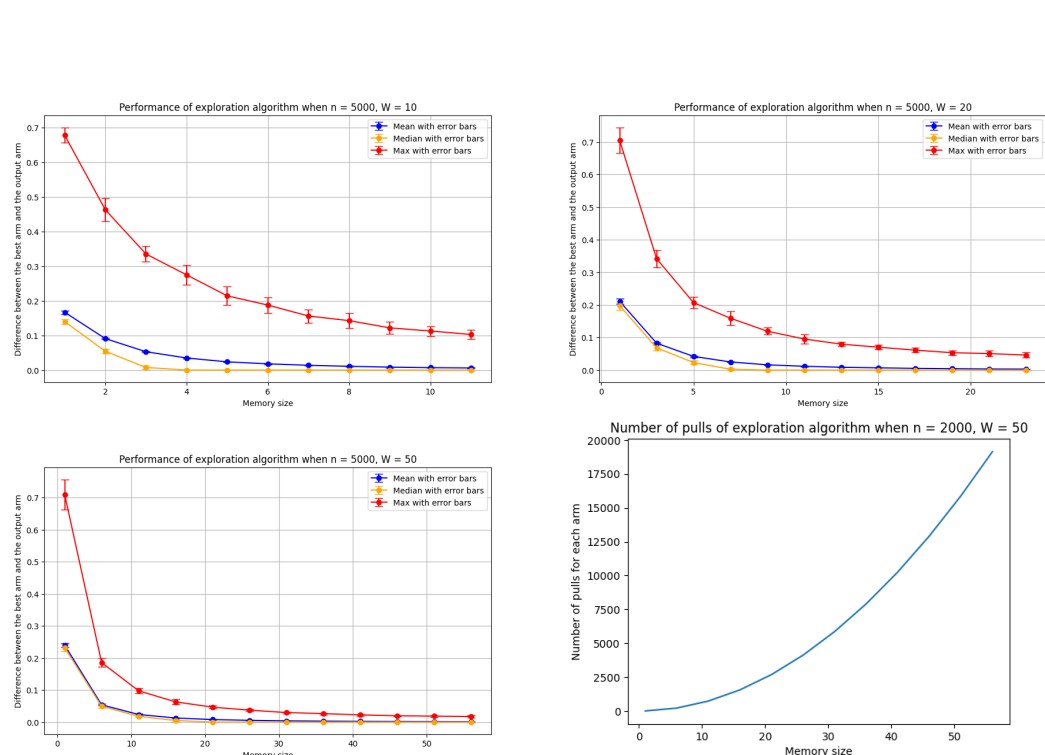

Figure 4: An illustration of the memory-quality trade-off in $\varepsilon$ exploration for $n = 5000$.

## G.2 EXPERIMENTAL RESULTS ON REGRET MINIMIZATION

We implement our $W$-memory algorithms for both the *everlasting regret* (the algorithm in Theorem 4) and *epoch-wise regret* (the algorithm in Theorem 3) cases. The purpose of the experiments is to show that the regret drops sharply once we have $W$ arm memory. To this end, we need to define how to proceed with the $W$-memory algorithms when we only have $o(W)$ arm memory. A natural approach is to simulate the reservoir sampling: after the memory is full, for each arriving arm, we toss a fair coin with bias $m/t$ for the $t$-th arriving arm to decide whether we admit the new arm to the memory (by uniformly at random discarding an arm existing in the memory).

**Experiments for regret with the everlasting best arm.** The experimental results for regret minimization with the everlasting best arm are shown as Figures 5 to 7. Here, we commit to any arm in the end if we do not have the best arm in the memory. With 10 independent runs of the algorithm, we report both the mean regret and the range of the regrets.

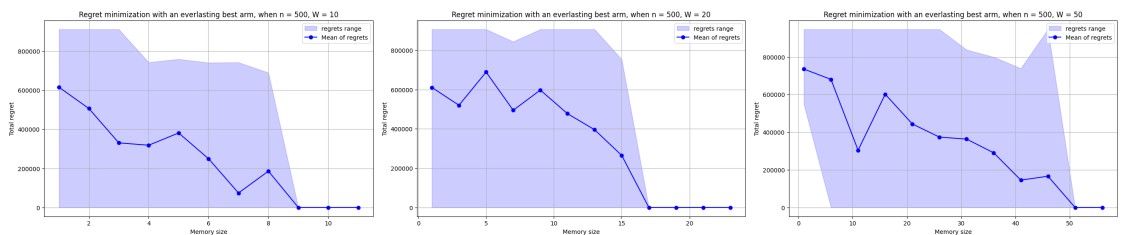

Figure 5: An illustration of the memory-regret trade-off for the everlasting best arm setting with $n = 500$.

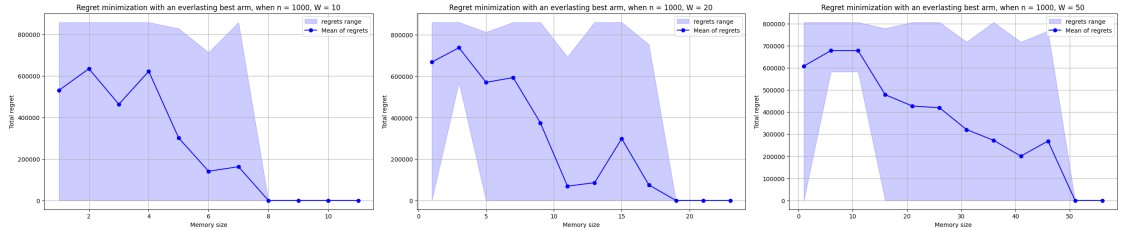

Figure 6: An illustration of the memory-regret trade-off for the everlasting best arm setting with $n = 1000$.

As we could observe in the figures, among the 10 executions of the algorithm, although the algorithm might get "lucky" with $o(W)$ memory, the range of the regret before reaching the $W$ memory is always wide, and the regret could always be high. On the other hand, after we have $W$ memory, we could easily identify the best arm and achieve 0 memory. The ranges observe a sharp drop at the $W$-memory point, which validates our theoretical results.

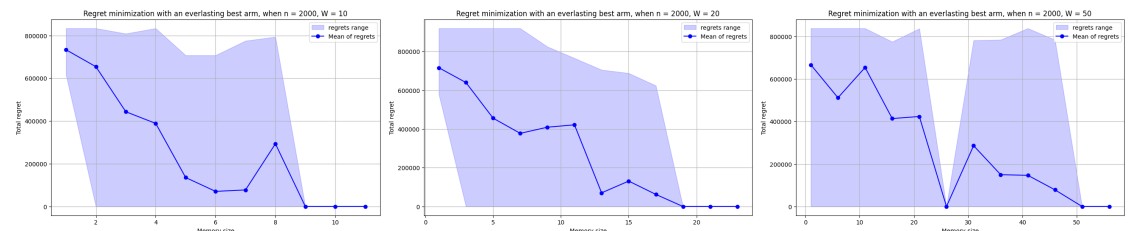

Figure 7: An illustration of the memory-regret trade-off for the everlasting best arm setting with $n = 2000$.

**Experiments for regret with the epoch-wise regret.** The experimental results for regret minimization in the epoch-wise regret setting are shown as Figures 8 to 10. If $m < W$, we will run UCB-based algorithms on the arms in the memory. Again, with 10 independent runs of the algorithm, we report both the mean and the range of the regrets.

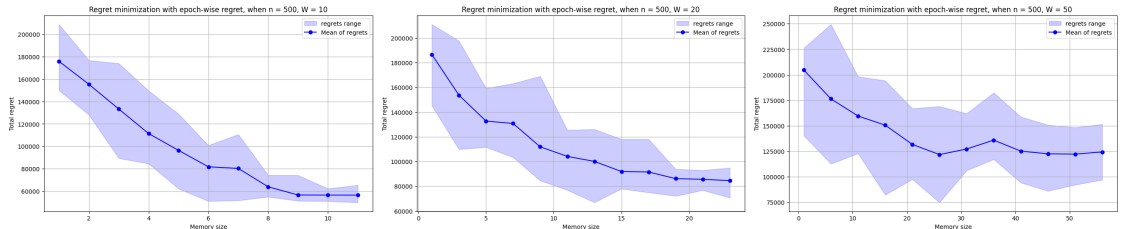

Figure 8: An illustration of the memory-regret trade-off for the epoch-wise regret setting with $n = 500$.

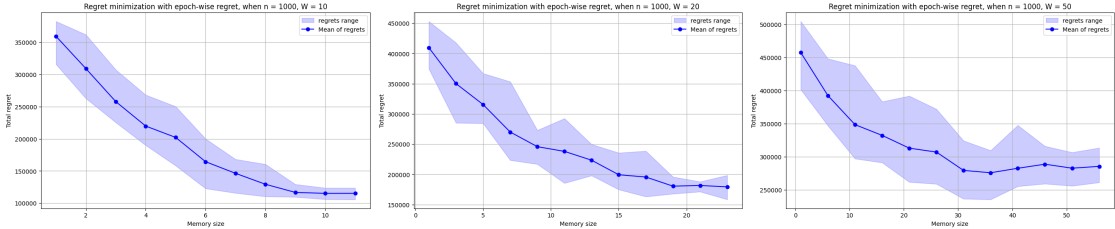

Figure 9: An illustration of the memory-regret trade-off for the epoch-wise regret setting with $n = 1000$.

From the figures, it could be observed that although the memory-regret trade-offs are smoother than in the case of the everlasting-regret setting, the trend still follows our result in Theorem 3. Furthermore, after reaching the memory of $W$ arms, the regret basically does not change with more memory, and the variances become much smaller.

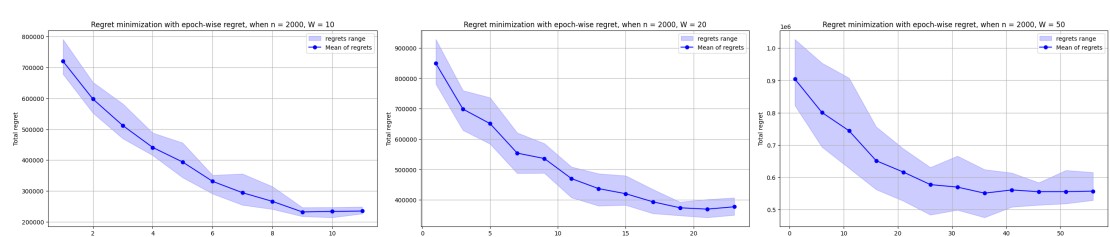

Figure 10: An illustration of the memory-regret trade-off for the epoch-wise regret setting with $n = 2000$.