# OpenReview forum: "Online Learning with Recency: Algorithms for Sliding-window Streaming Multi-armed Bandits"
_ICLR.cc/2026/Conference — Submitted to ICLR 2026_

### Official Review · Reviewer_hDQv · 2025-10-27

**Soundness:** 2
**Presentation:** 3
**Contribution:** 2
**Rating:** 4
**Confidence:** 4

**Summary:**

This paper studied a streaming multi-armed bandit problem. Instead of targeting a global objective, such as identifying the good arms or minimizing regret, the learner seeks to identify the good arms and minimize regrets in a sliding window containing $W$ arms. An exploration algorithm is proposed, and the regret minimization approach is developed based on it. Analytical results of the proposed algorithms are provided.

**Strengths:**

1. The paper is well written and easy to follow.
2. The theorems look reasonable, but I did not check the Appendix for details.
3. Experimental results are presented.

**Weaknesses:**

- No baselines are provided in the experiment section. Other algorithms, such as the track-and-stop and phase elimination, may be modified to address the sliding-window streaming MABs. Showing existing algorithms failing to achieve the local objective in the sliding window could highlight the significance of this work.

- Though the theoretical results make sense to me, the technical challenge is unclear. Since the mean rewards are assumed to be bounded by $[0,1]$, to determine the $\epsilon$-best arm, it is natural to divide the range into buckets of size $O(\epsilon)$. Also, the dependence of sample complexity on the window size $W$ seems straightforward. Intuitively, to achieve a miss detection rate $\leq \delta$, each arm can be sampled to till the empirical mean is $\epsilon$ close to the true mean with probability $1-\delta/W$.

- The regret minimization setup seems trivial to me, especially after the pure exploration algorihtm is established. One could also use other MAB algorithms, e.g., Thompson sampling or UCB, to minimize regret in each epoch. These algorithms can also be modified to be memory-efficient if $O(\epsilon T)$ regret is allowed.

-  The regret defined in separate epochs seems artificial, and the application is unclear. Why do existing models, such as arm-acquiring bandit [1], sleeping experts problem [2], mortal MABs [3], and nonstationary bandits, fail to address those scenarios? The paper could include the discussion.

[1] Whittle, Peter. "Arm-acquiring bandits." The Annals of Probability 9.2 (1981): 284-292.

[2] Robert Kleinberg, Alexandru Niculescu-Mizil, and Yogeshwer Sharma. Regret bounds for sleeping experts and bandits. Machine learning, 80(2):245–272, 2010.

[3] Chakrabarti, Deepayan, et al. "Mortal multi-armed bandits." Advances in neural information processing systems 21 (2008).

**Questions:**

Questions

- Could you explain the technical challenge for this problem?
- I think the phase elimination algorithm is a perfect fit for the sliding-window streaming MABs, since the bad arms are consistently eliminated from the candidate set. Why is it not mentioned in this paper?
- Could you give more example applications of the epoch-wise regret minimization?

Suggestions

The following suggestions are only for discussion. They do not need to be addressed.

- The problem could be more interesting if the problem-dependent sampling complexity (and regret bound) were studied. And I think the phase elimination could be a very good approach in this setup.

- The reverse version of the proposed problem might be more interesting. Given a fixed memory, what performance can be achieved? It is clear that the sliding-window characterization is a good one.

---

> ### Author Response · Authors · 2025-11-26
>
> We thank the reviewers for the careful review and insightful questions. On the other hand, we also believe there are some misunderstandings, and we want to respond as follows.
>
> > Though the theoretical results make sense to me, the technical challenge is unclear. Since the mean rewards are assumed to be bounded by $[0, 1]$, to determine the $\varepsilon$-best arm, it is natural to divide the range into buckets of size $O(\varepsilon)$. Also, the dependence of sample complexity on the window size $W$ seems straightforward. Intuitively, to achieve a miss detection rate $\delta$, each arm can be sampled to till the empirical mean $\varepsilon$ is close to the true mean with probability $1-\delta/W$.
> > Could you explain the technical challenge for this problem?
>
> We thank the reviewer for the insightful question. We remark that while the tools we used in analyzing the algorithms and lower bounds are standard, the technical challenges mainly lie in the *memory constraints* and *expiration of arms*.
>
> For the challenge of memory efficiency, note that the sliding-window setting is a strictly harder setting than the streaming multi-armed bandit setting by, e.g., Assadi and Wang, Exploration with Limited Memory: Streaming Algorithms for Coin Tossing, Noisy Comparisons, and Multi-Armed Bandits [STOC’20] and Jin, Huang, Tang, and Xiao, Optimal Streaming Algorithms for Multi-Armed Bandits [ICML’21]. Most bandit algorithms cannot be directly applied to the streaming or sliding-window setting with low memory.
>
> Furthermore, the streaming MABs algorithm, e.g., the algorithm in the AW [STOC’20] and JHTX [ICML’21], cannot be directly applied to the sliding-window setting. This is due to the expiration of arms: while the algorithms in the streaming papers can obtain good guarantees with respect to the overall best arm, it is easy to construct examples such that their guarantees in the sliding-window model are quite bad.
>
> Our algorithms for $\varepsilon$-exploration were developed aimed at the above context. We agree that the bucketing idea is natural and has been studied by various papers before. However, it is still nice to see that technically, the framework is compatible with the sliding-window model with a memory efficiency that is *independent of* $W$. Furthermore, it also takes some effort to figure out the right dependencies on the sample complexities (additional $\log{W}$ factor for weak exploration vs. $\log{n}$ factor for strong exploration).
>
> Finally, we want to remark that our lower-bound results are non-trivial and somewhat surprising. The algorithms in AW [STOC’20] and JHTX [ICML’21] have shown that by some clever algorithm design, we can find the overall best arm with high constant probability, optimal sample complexity, and the memory of one arm. In contrast, we proved that for the sliding-window setting, finding the exact best arm would require $\Omega(W)$ memory, no matter how many samples the algorithm uses.
>
> **Revision updates:** We have updated the introduction section by trimming the passage for contributions and results and adding more technical insights to the algorithms.
>
> > The regret minimization setup seems trivial to me, especially after the pure exploration algorihtm is established. One could also use other MAB algorithms, e.g., Thompson sampling or UCB, to minimize regret in each epoch. These algorithms can also be modified to be memory-efficient if $O(\varepsilon T)$ regret is allowed.
>
> We agree that these algorithms can be modified to be memory-efficient if $O(\varepsilon T)$ regret is permitted. However, $O(\varepsilon T)$ regret is usually considered very large. Usually, $\varepsilon$ values are considered as constants, and this will result in $\Omega(T)$ regret, which is asymptotically as bad as any pulling sequence.
>
> Similar to the exploration setting, the main technical challenges lie in the requirement of **space efficiency** and the exploration of arms. Due to the expiration problem, no existing algorithm can achieve sublinear regret ($o(T)$) in this model.
>
> It is true that the **algorithm** for our regret minimization basically stores all arms that have not expired, and runs the best offline regret minimization algorithm. The main technical challenge for regret minimization is to find a distribution of arms such that the expected regret on the distribution is high. To this end, we explored the ‘gradual decreasing mean’ distribution that forces any algorithm with sublinear regret to store $\Omega(W)$ arms.

---

> > ### Author Response · Authors · 2025-11-26
> >
> > > The regret defined in separate epochs seems artificial, and the application is unclear. Why do existing models, such as arm-acquiring bandit (1), sleeping experts problem (2), mortal MABs (3), and nonstationary bandits, fail to address those scenarios? The paper could include the discussion.
> >
> > We respectfully disagree with the statement that epoch-wise regret seems artificial.
> > Note that in our model, it is not immediately clear how to define regret due to the *expiration* of arms, i.e., the arms outside the $W$-sized window become invalid for considerations. Therefore, the “vanilla” definition of regret is not well-defined: since the algorithm can control the number of arm pulls before the window shifts, the regret's definition becomes a function of the algorithm. Therefore, we define regret in an epoch-wise manner such that the definition will be independent of the algorithm procedures.
> >
> > We also remark that reviewer hjY2 believes our notion of epoch-wise regret is well posed:
> > >> "Well-posed regret notion: Epoch-wise regret resolves the definitional pitfall of naive windowed regret and supports sharp lower/upper bounds."
> >
> > For application, consider movie recommendation systems. Here, each arm represents a movie, and each window corresponds to the period a movie is shown in theaters. The number of pulls, $T$, in regret minimization represents the advertisements shown to users. We want to focus these ads on movies likely to be popular and, ideally, always recommend the most relevant movies to users. This corresponds to the regret minimization problem in our model.
> >
> > > Why do existing models, such as arm-acquiring bandit [1], sleeping experts problem [2], mortal MABs [3], and nonstationary bandits, fail to address those scenarios? The paper could include the discussion.
> >
> > > I think the phase elimination algorithm is a perfect fit for the sliding-window streaming MABs, since the bad arms are consistently eliminated from the candidate set. Why is it not mentioned in this paper?
> >
> > We thank the reviewer for the suggestions for the algorithms. Nevertheless, we believe none of the algorithms mentioned in the review works for the sliding-window setting. There are two main issues to adapt these approaches to the sliding-window setting: memory efficiency and the expiration of arms.
> >
> > To begin with, none of the mentioned algorithms is known to have efficient *streaming* applications, leaving alone the sliding-window setting. (Please note that the sliding-window model is strictly more challenging than the streaming model.) Take the phase elimination algorithm as an example: the sample-efficient elimination algorithms are round-based, i.e., reading all arms before sampling all of them for some time. This inherently requires a large amount of memory, and the recent line of work in streaming multi-armed bandits, e.g., by Assadi and Wang, Exploration with Limited Memory: Streaming Algorithms for Coin Tossing, Noisy Comparisons, and Multi-Armed Bandits [STOC’20] and Jin, Huang, Tang, and Xiao, Optimal Streaming Algorithms for Multi-Armed Bandits [ICML’21], took considerable efforts to simulate elimination-based algorithms with a small amount of stored arms. Similar situations are true for arm-acquiring bandits, sleeping expert algorithms, mortal MABs, and nonstationary MABs – here, it is unclear whether these problems admit memory-efficient streaming algorithms (they are very interesting open problems to study for future research, though.). Therefore, it is hard to adapt these algorithms to the *sliding-window* model with memory constraints, which is a more restrictive model than streaming.
> >
> > The second challenge is the arm expiration in the sliding-window model: in our model, the best arm is only defined among the $W$ most recent, non-expired arms, not over all time. An arm may initially be worse than some other arms when it enters the stream, yet it could become the best in the future if all better arms expire and the subsequent arms perform even worse. Therefore, while many algorithms excel at eliminating bad arms to find the best arm over all time, they may fail in the sliding-window model, where any arm can be bad temporarily but become good later. Other algorithms that considered arm expiration in the same spirit, e.g., the mortal bandit algorithm, are not directly comparable to our setting since we already let the *last W arriving arms* be valid.
> >
> > **Revision update:** With all the above being said, we agree that the suggested algorithms are relevant to our problem. In the updated version of the paper, we have included a discussion about the relationship between our problem and the related problems (in a new appendix), and why algorithms in the memory-unconstrained setting cannot be directly applied to our model (the 'our techniques' section).

---

> > > ### Author Response · Authors · 2025-11-26
> > >
> > > > The problem could be more interesting if the problem-dependent sampling complexity (and regret bound) were studied. And I think the phase elimination could be a very good approach in this setup.
> > >
> > > We assume you mean the sample complexity and regret that are instance-dependent, i.e., with $\Delta_{[i]}$ as the parameters.
> > >
> > > To the best of our knowledge, the gap-dependent bounds are usually for the exploration of the *exact best* arm; however, in Theorem 1, we already proved that any algorithm that finds the *exact best* arm requires $\Omega(W)$ memory, which is not very interesting for the exploration problem.
> > >
> > > Also, the elimination-based algorithms are unlikely to succeed due to the lack of memory-efficient implementations.
> > >
> > > Whether we can get anything that is instance-dependent is an interesting direction to pursue, although we believe it goes beyond the scope of this paper.
> > >
> > > > The reverse version of the proposed problem might be more interesting. Given a fixed memory, what performance can be achieved? It is clear that the sliding-window characterization is a good one.
> > >
> > > We agree that the version of the problem with a fixed memory is interesting. For results in streaming and sliding-window MABs, some types of ‘sharp phase transition’ phenomenon could usually happen: below a certain memory, nothing can be done; if we go above some memory threshold, the problem suddenly starts to admit efficient algorithms. Our results already showed these types of properties, and the fixed-memory setting can be an interesting future direction to explore.

---

### Official Review · Reviewer_XSQq · 2025-10-31

**Soundness:** 3
**Presentation:** 2
**Contribution:** 2
**Rating:** 4
**Confidence:** 3

**Summary:**

The authors studied pure exploration and regret minimization in the sliding-window streaming multi-armed bandit setting, an extension of the steaming multi-armed bandits model where only the most recent W arms (sliding window) are considered valid. This formulation captures recency effects that arise in applications where underlying trends shift over time or where data retention is limited for privacy reasons. For the pure exploration problems, the authors show that identifying the exact best arm requires $\Omega(W)$ memory, while finding $\epsilon$-approximate best arm can be achieved with only $O(1/\epsilon)$ memory. For regret minimization, they introduce a new notion of sliding-window regret and establish a trade-off between memory and regret, demonstrating how performance degrades as memory constraints tighten.

**Strengths:**

This paper is the first to extend the sliding-window streaming model to the multi-armed bandit setting. For pure exploration problem, the authors established a upper bound for weak $\epsilon$-approximation and a lower bound for strong $\epsilon$-approximation that matches up to an extra $\log n$ factor. For the regret minimization problem, the paper introduces a new notion of sliding-window regret that is defined epoch-wise and does not depend on the arm pulls chosen by the algorithm. Under this refined definition, the authors prove a strong lower bound, showing that achieving a total regret smaller than $O(T/W^2)$ is impossible with $o(W)$ space.

**Weaknesses:**

A concern lies in the ambiguity of the scaling of the sliding window parameter $W$ which is a key component of the proposed model. For pure exploration, the lower bound is $\Omega(W)$, while in the later sections, $W$ appears to be treated as a constant. In fact, the scaling of all terms need to be made clear in the definitions and bounds. This is also the case for regret minimization problem. The lower and upper bound there do not seem to match, but the mismatch is not clearly explained, and the scales of the two bounds are not directly comparable. The paper would benefit from a clear and consistent specification of how $W$ and other parameters scale throughout the analysis.

The presentation of the paper could be improved. The main contributions and results are repeated verbosely, while the key derivations and methodological insights are presented too tersely. Given the somewhat incremental nature of the work, it would strengthen the paper to reduce redundancy and instead provide more discussion, intuition, and sketch proofs to clarify why the proposed results are nontrivial and how the algorithm brings methodological advances.

**Questions:**

1. In the motivation example of theatre visits, does the sliding windows correspond to a time period of 1-2 months and T the number of theatre visits? If so, T could be smaller than n-W+1 in this case. Could the authors clarify this interpretation?
2. For the strong $\epsilon$ pure exploration case, any insights into why using a larger pulling size alone can improve the bound alone helps match the lower bound?
3. What is the assumption for arm distributions? The experiments seem to rely on Bernoulli arms, which may be overly simplistic for modeling streaming settings.

---

> ### Author Response · Authors · 2025-11-26
>
> We thank the reviewers for the careful review and insightful questions. On the other hand, we also believe there are some misunderstandings, and we want to respond as follows.
>
> > A concern lies in the ambiguity of the scaling of the sliding window parameter $W$ which is a key component of the proposed model. For pure exploration, the lower bound is $\Omega (W)$, while in the later sections, $W$ appears to be treated as a constant. In fact, the scaling of all terms need to be made clear in the definitions and bounds. This is also the case for regret minimization problem. The lower and upper bound there do not seem to match, but the mismatch is not clearly explained, and the scales of the two bounds are not directly comparable. The paper would benefit from a clear and consistent specification of how $W$ and other parameters scale throughout the analysis.
>
> We want to emphasize that throughout our work, $W$ is always treated as a **large parameter** too large to be stored in memory, rather than a constant. Therefore, the upper and lower bounds of Result 2 scale with the parameter $W$ (as an input to the algorithm). On the other hand, our space upper bound in Result 1 is independent of $W$ – this is *not* because we treat $W$ as a constant, but rather a consequence of the design of our algorithm. We also remark that sliding-window algorithms with space bounds independent of $W$ are not uncommon: classical sliding-window algorithms, e.g., Braverman and Ostrovsky, Smooth Histograms for Sliding Windows [FOCS'07], can achieve space bounds that do *not* scale with $W$.
>
> Regarding the perceived mismatch between the upper and lower bounds, we want to clarify the settings for these bounds.
> -  In Theorem 1, we show that the lower bound for the memory required to solve pure exploration, aimed at identifying the best arm, is $\Omega(W)$. Thus, it is (asymptotically) impossible to outperform a trivial algorithm that stores all non-expired arms.
> - For the slightly different $\varepsilon$-exploration problem, which aims to find an $\varepsilon$-best arm, we develop an algorithm with a memory bound of $O(\frac{1}{\varepsilon})$, which does not depend on $W$ (Theorem 2).
> - The bounds for (strong) $\varepsilon$-exploration are also tight, as shown by Lemmas 4.1 and 4.2.
>
> The main conceptual message of the bounds is as follows: for the exact pure exploration setting, nothing can be done other than the trivial algorithm with $\Theta(W)$ memory. However, if we relax the setting to $\varepsilon$-exploration, it is suddenly possible to obtain algorithms with a memory of $O(\frac{1}{\varepsilon})$ arms.
>
> Our regret upper and lower bounds follow the same logic: with $o(W)$ memory, nothing can be done, and we will need to pay $\Omega(T/W^2)$ regret. However, if we slightly increase the memory to $\Theta(W)$, we can suddenly get a regret that is optimal on $T$ ($O(\sqrt{T})$ regret).
>
> **Revision updates:** we have emphasized the fact that $W$ is always treated as a parameter, clarified the different settings for our bounds, and added a table to illustrate the roles of the bounds.
>
> > The presentation of the paper could be improved. The main contributions and results are repeated verbosely, while the key derivations and methodological insights are presented too tersely. Given the somewhat incremental nature of the work, it would strengthen the paper to reduce redundancy and instead provide more discussion, intuition, and sketch proofs to clarify why the proposed results are nontrivial and how the algorithm brings methodological advances.
>
> We thank the reviewer for their writing suggestions and have incorporated these changes into the revised version of the paper. However, we do not consider our results incremental, as this is the first set of results applicable to sliding-window streaming bandits.
>
> **Revision updates:** We have updated the introduction section by trimming the passage for contributions and results and adding more technical insights to the algorithms. We have also added an image table to clarify the bounds.
>
> > In the motivation example of theatre visits, does the sliding windows correspond to a time period of 1-2 months and T the number of theatre visits? If so, T could be smaller than n-W+1 in this case. Could the authors clarify this interpretation?
>
> For the example of theater visits, the 'bandits' can be considered the movies shown in theaters across the United States over the past 40 years. The 'window' can represent a time period of 1-2 months, and 'T' can refer to the total number of theater visits over all time. Therefore, we have approximately n = 15,000, W = 60, and T = 40,000,000,000. These numbers are only rough estimates and may not be highly accurate, but they illustrate that in this example, $T \gg n \gg W$.

---

> > ### Author Response · Authors · 2025-11-26
> >
> > > For the strong pure $\varepsilon$-exploration case, any insights into why using a larger pulling size alone can improve the bound alone helps match the lower bound?
> >
> > The difference between strong and weak $\varepsilon$-exploration is that the weak version requires a success rate greater than $1-\delta$ at any given time, whereas the strong version demands this rate over the entire time horizon. Intuitively, if we increase the number of arm pulls, our estimation confidence is improved, leading to better success probability.  In the strong $\varepsilon$-exploration, the error probabilities accumulate over all $n$ rounds. By increasing the number of pulls, we can reduce the error to $O\left(\frac{\delta}{n}\right)$ to overcome the requirement in the union bound among $n$ rounds.
> >
> > > What is the assumption for arm distributions? The experiments seem to rely on Bernoulli arms, which may be overly simplistic for modeling streaming settings.
> >
> > All of our algorithms apply to distributions supported between $0$ and $1$. More generally, this can be extended to assume that each arm follows a sub-Gaussian distribution up to a rescaling factor—a standard generalization in prior MAB research. On the other hand, all of our lower bounds use Bernoulli distributions. Note that this only makes our results **stronger**: if a lower bound applies to Bernoulli distributions, it automatically applies to all sub-Gaussian distributions. In our experiments, we use Bernoulli arms—not because we assume an overly simplistic Bernoulli distribution for theory, but to facilitate easier implementation.
> >
> > On the other hand, we do not make any assumption about the arrival order of the stream. In other words, all our algorithms work even if the arrival order of the stream is adversarial.

---

### Official Review · Reviewer_hjY2 · 2025-11-01

**Soundness:** 3
**Presentation:** 3
**Contribution:** 3
**Rating:** 6
**Confidence:** 3

**Summary:**

The paper formalizes a sliding-window streaming MAB setting where arms arrive once in a stream and, at time $t$, only the most recent $W$ arrivals are considered \emph{valid}. An algorithm may store a subset of past arms (subject to a memory budget) and, upon each arrival, decide whether to pull, store, or discard. The work studies two tasks:


\textbf{Pure exploration.} The authors show that \emph{exact} best-arm identification within the active window requires $\Omega(W)$ memory in a single pass (even with many pulls). In contrast, $\varepsilon$-best identification is achievable with space $O(1/\varepsilon)$ via a simple \textsc{BUCKET} scheme that partitions rewards into $O(1/\varepsilon)$ bins and keeps the newest representative per bin. Sampling complexity is $O\!\big(\tfrac{n}{\varepsilon^2}\log \tfrac{W}{\delta}\big)$ in the weak setting and $O\!\big(\tfrac{n}{\varepsilon^2}\log \tfrac{n}{\delta}\big)$ in the strong setting.

\textbf{Regret minimization.} Because naive regret is ill-posed (the learner can shift pulls across time), the paper introduces \emph{epoch-wise regret}: split the horizon into $n-W+1$ epochs and enforce exactly $T/(n-W+1)$ pulls per epoch. Under this notion, $\Omega(W)$ memory is necessary to achieve $o(T)$ regret; with $O(W)$ memory, the authors give an algorithm with regret $O\!\big(\sqrt{W\,(n-W)\,T}\big)$ (matching centralized $O(\sqrt{nT})$ when $W$ is constant).


Experiments on synthetic Bernoulli streams validate: (i) a smooth memory--quality trade-off for $\varepsilon$-exploration and (ii) a sharp drop in regret once memory reaches $W$.

**Strengths:**

1. Crisp model for recency: The sliding-window abstraction makes the ``recent-only'' constraint explicit and separates validity of arms from memory limits, enabling clean space/sample guarantees.

2. Fundamental hardness: The $\Omega(W)$ space lower bound for exact best arm (single pass) is clear and compelling, isolating an intrinsic barrier tied to the window size.

3. Practical approximate exploration: The \textsc{BUCKET} algorithm attains space $O(1/\varepsilon)$ and near-optimal sample complexity using standard concentration, with an implementation that is simple and streaming-friendly.

4. Well-posed regret notion: Epoch-wise regret resolves the definitional pitfall of naive windowed regret and supports sharp lower/upper bounds.

5. Tight memory--regret frontier. The results pinpoint a phase transition: $o(T)$ regret needs $\Omega(W)$ memory, while $O(W)$ memory suffices for $O\!\big(\sqrt{W(n-W)T}\big)$ regret; this situates the setting relative to classical $O(\sqrt{nT})$ bounds.

6. Empirical corroboration. Simulations reproduce the predicted phase transition (near memory \(W\)) and demonstrate the expected behavior of the \textsc{BUCKET} scheme.

**Weaknesses:**

1.  Motivation vs. storage model (expired arms can be kept): The model permits retaining any past arms in memory even after they leave the valid window. For privacy/retention-driven applications---a stated motivation---expired data would typically need to be deleted, likely tightening the achievable space/sample trade-offs. A variant that forbids storing expired arms (or charges for it) would better align with such use cases and reveal which guarantees survive under stricter retention.

2. Fixed pulls per epoch may limit policy expressiveness: Epoch-wise regret fixes exactly $T/(n-W+1)$ pulls in each epoch to avoid algorithm-dependent benchmarks. This is mathematically clean, but constrains adaptive allocation of effort across epochs (e.g., heavier exploration right after a suspected shift). A relaxed notion with bounded per-epoch variability could preserve well-posedness while covering common operational practices.

3. Experimental scope is narrow and fully synthetic: Experiments consider Bernoulli rewards and do not include head-to-head comparisons with strong non-windowed streaming heuristics (e.g., sliding-window UCB/TS used naively) nor real/semi-synthetic traces where recency is known to matter. Broader baselines and stress tests (varying $W$, non-stationary gaps) would better establish practical benefits.

4. Worst-case lower bounds may overstate typical memory needs: The $\Omega(W)$ space lower bound for exact best-arm relies on a descending-means construction (Yao-style). In benign instances with large gaps, exact identification could be feasible with $o(W)$ memory. Presenting instance-dependent or average-case refinements would contextualize the worst-case message.

5. \textsc{BUCKET} is gap-agnostic and can be conservative:
 The algorithm partitions $[0,1]$ into $O(1/\varepsilon)$ bins and uses uniform per-arm sampling $s=\Theta(\varepsilon^{-2}\log(\cdot))$, independent of realized gaps. Gap-adaptive policies (e.g., racing/successive elimination adapted to the window) could reduce pulls substantially when many arms are clearly suboptimal. A comparison or ablation would help practitioners gauge efficiency.

6. Heavy-tailed rewards are not addressed:
 Analyses assume bounded or sub-Gaussian rewards. Many streaming logs are heavy-tailed; clarifying whether the results extend with robust estimators (median-of-means, Catoni) would widen applicability.

7. Limited practical guidance for tuning $\varepsilon, W$, and memory:
While the asymptotic rates are clear, practitioners will ask: given a memory cap $m$, how to pick $\varepsilon$ (or bucket count) and expected sample budget to meet a target error/regret? A concise design table or rule-of-thumb derived from the theorems would increase usability.

8. Claims about prior methods would benefit from direct baselines:
 The text argues earlier streaming MAB methods can output stale/invalid arms in recency-sensitive settings. A head-to-head empirical comparison (even if those baselines are naively adapted) would make this gap concrete and quantify the advantage.

**Questions:**

Q.1. Privacy-consistent retention: The paper’s motivation includes data-retention and privacy considerations, yet the formal model allows the learner to retain arms even after they exit the valid window. This creates a gap between motivation and assumptions. Please clarify which results (e.g., the $\Omega(W)$ space lower bound for exact best-arm and the $O(1/\varepsilon)$-space $\varepsilon$-exploration guarantees) continue to hold if expired arms must be dropped immediately, and which ones would need to change. A short lemma or an experiment under a “no-storage-after-expiration’’ constraint would make the scope precise.

Q.2. Regret with bounded per-epoch variability: Epoch-wise regret fixes exactly $T/(n{-}W{+}1)$ pulls per epoch to avoid an algorithm-dependent comparator. While clean, this rigidity may not reflect realistic allocation patterns where effort is rebalanced after suspected shifts. It would strengthen the contribution to discuss a bounded-variation alternative and whether the main lower/upper bounds survive or fail under that relaxation; a succinct counterexample would be equally valuable.

Q.3. Gap-adaptive exploration under windows: The proposed \textsc{BUCKET} scheme is gap-agnostic: it fixes binning and sample counts regardless of realized difficulty. Many deployments exhibit large gaps where racing/successive-elimination saves pulls dramatically. Please consider a window-aware gap-adaptive baseline and report whether similar $O(1/\varepsilon)$ space can be preserved while reducing samples on easy instances; an ablation would clarify the practical efficiency frontier.

Q.4. Heavy-tailed rewards: The analysis relies on bounded or sub-Gaussian rewards, which can be violated by bursty or long-tailed streams. A brief discussion of robust mean estimators (median-of-means, Catoni) and the conditions under which the stated space/sample orders continue to hold would broaden applicability. Even a proof sketch or an appendix lemma would suffice.

Q.5. Instance-dependent or average-case lower bounds: The $\Omega(W)$ space lower bound for exact best-arm is worst-case (descending-means style). Practitioners will care about typical regimes where within-window gaps are large. Complementing the worst-case with instance-dependent lower bounds (in terms of the gap structure) or with empirical evidence that exact best-arm often needs $o(W)$ memory in benign instances would contextualize the hardness claim.

Q.6. Baselines and stress tests: Practical relevance hinges on performance against strong streaming heuristics under a variety of conditions. Including sliding-window UCB/TS and reservoir-like policies, and stress-testing across $W$, $n$, and non-stationary gap profiles would help map out win/lose regimes and sensitivity. This would also illuminate whether the theoretical phase transition near memory $W$ is robust beyond the current synthetic setups.

Q.7. Design guidance for practitioners: The theorems give clear asymptotic rates; translating them into a simple recipe would aid adoption. A small table that maps a memory budget $m$ to a recommended $\varepsilon$, bucket count, and expected error/regret (with one worked example) would make the results immediately actionable for engineering teams.

Q.8. Effect of retaining expired arms: To isolate the contribution of retention, an ablation across three regimes—(i) free retention of expired arms, (ii) memory-penalized retention, and (iii) no retention—would quantify how much of the observed gains stem from storing expired items. Reporting quality/regret versus memory in each regime would make this trade-off transparent.

Q.9. Time/space complexity at stream scale: Streaming deployments are latency-sensitive. Reporting per-arrival processing time and memory footprints, and how they scale with $W$, $n$, and the number of buckets, would help readers judge deployability and identify bottlenecks or easy optimizations.

Q.10. Relation to non-stationary bandits with persistent arms: A concise subsection clarifying the conceptual difference between “recency-validity over arms with single-pass space constraints’’ (this work) and “recency over rewards for persistent arms’’ (standard drifting-mean methods) would situate the contribution and indicate which techniques from that literature can or cannot be transplanted here.

---

> ### Author Response · Authors · 2025-11-26
>
> We thank the reviewer for the careful review, the positive evaluation, and the insightful questions. We answer your questions as follows.
>
> > Motivation vs. storage model (expired arms can be kept) (**W1**) and Privacy-consistent retention (**Q1**)
>
> Thank you for the observation. While our definition allows expired arms to be stored in the memory, none of our algorithms or lower bounds actually use this property (except the *additional* setting with the everlasting best arm we discussed in Appendix D). For the $\varepsilon$-exploration algorithm (Algorithm 1), we explicitly stated that we discard all stored arms that have expired. The same holds for the regret-minimization algorithm, where we only perform regret minimization on the current window of valid arms. Finally, for the lower bound, we note that any algorithm that stores expired arms can be simulated by an algorithm that does *not* store expired arms with the same or smaller sample and memory complexity. Therefore, our lower bounds remain true if expired arms are not allowed to be kept in the memory.
>
> **Revision updates:** We added a remark after the model definition to discuss the unnecessary role of storing expired arms and its relationship with privacy retention.
>
> > Fixed pulls per epoch may limit policy expressiveness; mathematically clean, but constrains adaptive allocation of effort across epochs (**W2** and **Q2**)
>
> If we let window $i$ to have $T_i$ samples, with $W$ memory, we can get $\sum_{i} O(\sqrt{W T_i})$ regret such that $\sum_{i} T_i = T$. The parameter would make the bound quite much less informative, though. Our lower bound remains true in the case that, e.g., we concentrate all the samples on one of the windows $[k\cdot 2W+1, k\cdot 2W+W]$.
>
> **Revision updates:** We added a remark in Appendix D (previously Appendix C) to discuss additional settings of our results where the number of pulls in each epoch can vary.
>
> > Experimental scope is narrow and fully synthetic (**W3**)
>
> We agree with the reviewer that the current experiments are still preliminary and use only synthetic datasets. The primary goal of this paper is to establish a fundamental theoretical understanding of sliding-window MABs, rather than providing a comprehensive empirical analysis of the algorithms in this regime.
>
> Furthermore, we remark that the entire area of streaming/sliding-window MABs is still in the mostly theoretical arena, and existing work with experiments, e.g., Karpov and Wang, “Nearly Tight Bounds for Exploration in Streaming Multi-armed Bandits with Known Optimality Gap” [AAAI’25], also used only synthetic datasets. Getting results for real-world datasets for sliding-window MABs faces additional hurdles (see our responses to Q9), and we leave that as a future direction to explore.
>
> > Lower bounds are only for the worst-case; average-case performances (**W4** and **Q5**)
>
> We agree that the current lower bounds are only for the worst case. For streaming and sliding-window MABs, it is not always clear how to define “benign instances”: if the ordering of the arms is, e.g., increasing mean, the problem becomes trivial since the algorithm can simply output the arriving arm at every time. Random-order arrival might be an interesting setting, and we leave this direction for future exploration.
>
> > The bucket-based algorithm is gap-agnostic and can be conservative (**W5** and **Q3**)
>
> We agree with the reviewer that the bucketing strategy used in the paper is gap-agnostic. Nevertheless, we remark that algorithms for *$\varepsilon$-exploration* have to be gap-agnostic. The gap-dependent bounds are usually for the exploration of the *exact best* arm; however, in Theorem 1, we already proved that any algorithm that finds the *exact best* arm requires $\Omega(W)$ memory, which is not very interesting for the exploration problem.
>
> For the successive elimination algorithm, we can particularly argue that it does not admit streaming algorithms with low memory. The sample-efficient elimination algorithms are round-based, i.e., reading all arms before sampling all of them for some time. This inherently requires a large amount of memory, and the recent line of work in streaming multi-armed bandits, e.g., by Assadi and Wang, Exploration with Limited Memory: Streaming Algorithms for Coin Tossing, Noisy Comparisons, and Multi-Armed Bandits [STOC’20], and Jin, Huang, Tang, and Xiao, Optimal Streaming Algorithms for Multi-Armed Bandits [ICML’21], took considerable efforts to simulate elimination-based algorithms with a small amount of stored arms. However, these simulations are unlikely to work in the sliding-window setting.

---

> ### Author Response · Authors · 2025-11-26
>
> > Heavy-tailed rewards are not addressed (**W6** and **Q4**)
>
> We agree that we need sub-Gaussian distributions for the algorithms to work. To the best of our knowledge, there is no known work for heavy-tail bandits even in the streaming MABs model. While we agree that it is an interesting direction, the problem has gone much further than the scope of this research.
>
> > Limited practical guidance for tuning parameters $W$, $n$, and $\varepsilon$ (**W7** and **Q6, Q7**)
>
> First, we remark that the parameters $W$ and $n$ are functions of the input instance, not parameters that the algorithm designer/practitioner could tune. For the memory $m$ and the value of $\varepsilon$, the theory says that we should, in general, have $m=O(1/\varepsilon)$, and we believe this is clear enough without a table. The exact optimal constant would require algorithm engineering, and since the primary goal of this paper is to establish a fundamental theoretical understanding of sliding-window MABs, we leave that as a direction to pursue for future research.
>
> > Empirical comparisons with baselines (**W8**)
>
> For the exploration applications, it is unclear how to define the “error” between the output arm and the best arm in the window. Should we simply output the mean of the best arm as the error if the returned arm is not even valid? If that’s the case, the curve might be a bit weird in the sense that increasing samples cannot improve the performance. If the reviewer wants to see a plot that accounts for the error in this way, please let us know, and we can conduct additional experiments.
>
> **Revision updates:** We added a few sentences to discuss the difficulty of accounting for errors in running the vanilla streaming MABs algorithm on sliding-window instances.
>
> > Effect of retaining expired arms: To isolate the contribution of retention, an ablation across three regimes—(i) free retention of expired arms, (ii) memory-penalized retention, and (iii) no retention—would quantify how much of the observed gains stem from storing expired items. Reporting quality/regret versus memory in each regime would make this trade-off transparent. (**Q8**)
>
> As we have discussed in our response to **W1**, none of our upper and lower bounds actually use the retention of expired arms. Therefore, we do not have any trade-off if we disallow the retention of expired arms.
>
> > Time/space complexity at stream scale (**Q9**)
>
> The question is actually related to the broader context of implementing streaming algorithms in practice. We agree that streaming deployments are latency-sensitive. For our dataset, since it is fully synthetic, the processing time and latency are negligible. However, to implement our results in real-world large-scale streaming applications, it will take considerable resources and effort, to the extent that it will be another paper.
>
> Let us elaborate: Building streaming systems that are efficient and with low latency is an active research venue in systems. For instance, for the graph streaming model, the connected components algorithm was designed by Ahn et al. [SODA’12]. Nevertheless, it took much algorithm engineering and system optimization efforts to actually scale the system to large inputs (Tench et al., [SIGMOD’22], ‘GraphZeppelin’). Therefore, we believe the optimization of the system latency and processing time for the streaming and sliding-window MABs is an interesting problem in its own, and pursuing that will go beyond the scope of this paper.
>
> > Relation to non-stationary bandits with persistent arms (**Q10**)
>
> We thank the reviewer for pointing out the similar settings in non-stationary bandits (we noticed that reviewer hDQv also mentioned it).
>
> **Revision update:** In the updated version of the paper, we have included a discussion about the relationship between our problem and the related problems, e.g., nonstationary bandits, and why algorithms in the memory-unconstrained setting cannot be directly applied to our model (in ‘our techniques’ and a new appendix section due to space limitations).

---

### Author Response · Authors · 2025-12-03
**Author Final Summary**

We appreciate the reviewers for their careful review and insightful questions. We have addressed their concerns and clarified some misunderstandings in our responses to their comments. Below is a brief summary of the revisions made to our paper.

We have improved the overall presentation of the paper. The introduction section has been updated to include a more concise discussion of our contributions and results, as well as additional technical insights about the algorithms. We have emphasized that $ W $ is consistently treated as a parameter, clarified the different settings for our bounds, and included a table to illustrate the roles of these bounds. We also discuss the challenges of defining a well-defined regret minimization more thoroughly, highlighting our contribution in successfully defining regret minimization within the sliding window model.

In Section 2, we note the unnecessary role of storing expired arms and its implications for privacy retention, clarifying that whether or not we store expired arms does not affect the results of our research.

In Section 6, we have added a few sentences to discuss the challenges associated with accounting for errors when running the vanilla streaming MAB algorithm on sliding-window instances.

In Appendix B, we include a discussion about the relationship between our problem and related issues, such as nonstationary bandits, and explain why algorithms designed for the memory-unconstrained setting cannot be directly applied to our model.

In Appendix D, we added a remark addressing additional settings of our results where the number of pulls in each epoch may vary.

We believe our responses to the reviewers adequately address all their questions, and the revisions made to the paper further clarify their concerns.

---

### Meta-Review · Area_Chair_VZ4K · 2025-12-11

**Summary:**

The main concerns about this paper is:

i) Unclear motivation, particularly regarding the rationale behind the proposed regret definition.

ii) Lack of instance-dependent regret for the $\epsilon$-approximate pure-exploration case.

iii) The experiments are conducted primarily on synthetic datasets and do not include comparisons with existing baselines.

**Reviewer Concerns:**

I believe these three concerns are not fully addressed.

First, regarding the definition of regret, the use of uniformly partitioning the horizon into $n−W+1$ phases does feel artificial. The authors motivate this problem setting by arguing that data may "expire". However, the timing of expiration should naturally be an instance-specific property. For example, one could consider a total horizon $T$ where each arm $i$ is only available from $T_{i}^{start}$ to $T_{i}^{end}$, and these availability intervals form part of the problem instance. In my opinion, such a formulation aligns more closely with the applications discussed in the paper and would yield a more principled motivation for the regret definition.

Second, the authors state that no $\Delta$-dependent regret bound is provided for the $\epsilon$-approximate pure-exploration setting. However, by analogy with standard approximate best-arm identification results, it seems plausible that a $\Delta$-dependent sample-complexity bound on the order of $\tilde{O}(1/max(\epsilon^2,\Delta^2))$ could be achievable. At the very least, this potential direction and its relation to existing results could be discussed more.

**Reviewer Scores:**

The main concerns are not addressed, so I think reviewer hjY2 and hDQv will not change their scores. Regarding reviewer X5Qq, the questions he raised receive reasonably convincing answers in the rebuttal, so an increase of one point in his overall score seems possible.

---

### Decision · Program_Chairs · 2026-01-26

Reject